# Ambient carbonaceous aerosol levels in Cyprus and the role of pollution transport from the Middle East.

Aliki Christodoulou[1,2], Iasonas Stavroulas[1,3], Mihalis Vrekoussis[1,4,5], Maximillien Desservettaz[1], Michael Pikridas[1], Elie Bimenyimana[1], Jonilda Kushta[1], Matic Ivančič[6], Martin Rigler[6], Philippe Goloub[7], Konstantina Oikonomou[1], Roland Sarda-Estève[8], Chrysanthos Savvides[9], Charbel Afif[1,10], Nikos Mihalopoulos[1,3], Stéphane Sauvage[2] and Jean Sciare[1]

[1]Climate and Atmosphere Research Center (CARE-C), the Cyprus Institute, Nicosia, 2121, Cyprus
[2]IMT Nord Europe, Institut Mines-Télécom, Univ. Lille, Centre for Energy and Environment, 59000 Lille, France
[3]Institute for Environmental Research and Sustainable Development, National Observatory of Athens, Athens, Greece
[4]Institute of Environmental Physics and Remote Sensing (IUP), University of Bremen, Germany
[5]Center of Marine Environmental Sciences (MARUM), University of Bremen, Germany
[6]Aerosol d.o.o., Research & Development Department, Kamniška 39a, SI-1000 Ljubljana, Slovenia
[7]University of Lille, CNRS, LOA – Laboratoire d'Optique Atmosphérique, Lille, 59000, France
[8]Laboratoire des Sciences du Climat et de l'Environnement (LSCE), CNRS-CEA-UVSQ, Gif-sur-Yvette, France
[9]Ministry of Labour and Social Insurance, Department of Labour Inspection (DLI), Nicosia, Cyprus
[10]Emissions, Measurements, and Modeling of the Atmosphere (EMMA) Laboratory, CAR, Faculty of Sciences, Saint Joseph University, Beirut, Lebanon

*Correspondence to*: Aliki Christodoulou (a.christodoulou@cyi.ac.cy) and Jean Sciare (j.sciare@cyi.ac.cy)

**Abstract.** The geographical origin and source apportionment of submicron carbonaceous aerosols (organic aerosols, OA, and black carbon, BC) have been investigated here for the first time deploying high-time resolution measurements at an urban background site of Nicosia, the capital city of Cyprus, in the Eastern Mediterranean. This study covers a half-year period, encompassing both the cold and warm periods with continuous observations of the physical and chemical properties of $PM_1$ performed with an Aerosol Chemical Speciation monitor (ACSM), an Aethalometer, accompanied by a suite of various ancillary off and on-line measurements. Carbonaceous aerosols were dominant during both seasons (cold and warm periods), with a contribution of 57% and 48% to $PM_1$, respectively, and exhibited recurrent intense night-time peaks (>20-30 µg m$^{-3}$) during the cold period associated with local domestic heating. The findings of this study show that high concentrations of sulfate (close to 3 µg m$^{-3}$) were continuously recorded, standing among the highest ever reported for Europe and originating from the Middle East region.

Source apportionment of the OA and BC fractions was performed using the Positive Matrix Factorization (PMF) approach and the combination of two models (aethalometer model and multilinear regression), respectively. Our study revealed elevated hydrocarbon-like organic aerosol (HOA) concentrations in Nicosia (among the highest reported for a European urban background site), originating from a mixture of local and regional fossil-fuel combustion sources. Although air masses from the Middle East had a low occurrence and were observed mostly during the cold period, they were shown to strongly affect the mean concentrations levels of BC and OA in Nicosia during both seasons. Overall, the present study brings to our attention the need to further characterize primary and secondary carbonaceous aerosols in the Middle East; an undersampled region characterized by continuously increasing fossil fuel (oil and gas) emissions and extreme environmental conditions, which can contribute to photochemical aging.

## 1. Introduction

At the crossroads of three continents (Europe, Africa, Asia), the Eastern Mediterranean and Middle East (EMME) region faces many challenges, such as rapid population growth – with its currently 400 million inhabitants – as well as political and socio-economic instabilities. Environmental conditions in the region are exceptional, with the two largest deserts worldwide (Sahara

and Arabian) being among the most water scarce ecosystems on the planet (Terink et al., 2013). Climate change in this region
is extraordinarily rapid; summer temperatures, in particular, are increasing by more than twice the global mean rate (Lelieveld
et al., 2014), with significant impacts, especially in urban areas (Mouzourides et al., 2015). While aerosol mass loadings over
the EMME are dominated by desert dust, concentrations of fine particles due to anthropogenic emissions are also high (Basart
et al., 2009) and will likely increase with continued population growth (Pozzer et al., 2012), making anthropogenic pollution
in the area a leading health risk and an important climate forcer (Osipov et al., 2022).
Based on modelling studies, it has been also concluded that the EMME is characterized by highly favourable conditions for
photochemical smog and ozone ($O_3$) formation leading to air quality standards being drastically exceeded (Lelieveld et al.,
2014; Zanis et al., 2014). These enhanced concentrations of fine particulates and ozone have major human health implications,
contributing to premature mortality (Giannadaki et al., 2014; Lelieveld et al., 2015), which may be further exacerbated by the
effects of heatwaves occurring during summer within the EMME region (Zittis et al., 2022).
Although data from satellite observations of $NO_2$ and $SO_2$ has revealed strong air pollution trends in the Middle East since
2010 (Lelieveld et al., 2015a), many pollution sources are still missing in emission inventories (Mclinden et al., 2016). Thus,
there is a current lack of a regional approach to characterize air pollution, with in-situ observation being insufficient,
unavailable, or of low quality (Kadygrov et al., 2015; Ricaud et al., 2018; Paris et al., 2021), limiting the possibility to reduce
uncertainties in regional emission inventories and implement efficient abatement strategies.
Significant efforts have been put forward in recent years to characterize the atmospheric composition in-situ over Cyprus, a
central location of the EMME region (e.g., Kleanthous et al., 2014; Debevec et al., 2017 and 2018; Pikridas et al., 2018; Dada
et al., 2020; Baalbaki et al., 2021; Vrekoussis et al., 2022). In-situ ground-based PM observations have clearly shown that
contributions of dust to $PM_{10}$ over Cyprus are among the highest for the entire Mediterranean basin (Querol et al., 2009; Pey
et al., 2013; Kleanthous et al., 2014; Pikridas et al., 2018; Achilleos et al., 2020), during dust storm events, leading to increased
hospitalization, particularly attributed to cardiovascular-related diseases (Middleton et al., 2008; Tsangari et al., 2016) and
short-term effects associated with daily mortality (Neophytou et al., 2013). These high levels of regional particulate matter are
responsible for exceedances in $PM_{10}$ EU limits in major Cypriot cities (Querol et al., 2009). Past studies on PM trends and
sources highlighted the important contribution of local (urban) emissions to $PM_{10}$ (Achilleos et al., 2014; Pikridas et al., 2018)
but also showed a predominant regional pattern for $PM_{2.5}$ with a major contribution of sulfur-rich sources (Achilleos et al.,
2016). Based on 17 years of continuous observations of reactive gases in Cyprus, Vrekoussis et al. (2022) further confirmed
the major contribution of long-range transport (incl. Middle East) in the observed concentration levels of carbon monoxide
(CO) and sulfur dioxide ($SO_2$), two tracers of combustion sources.
Those studies have highlighted the unique location of Cyprus as a receptor site of major regional pollution hotspots, making
the island one of the most polluted EU member states in terms of PM and $O_3$ concentrations, the only one impacted by long-
range transport of poorly-regulated air pollutants originating from Middle East countries. However, few studies are currently
available to assess the contribution of regional anthropogenic emissions to PM levels in Cyprus. The filter-based chemical
speciation study reported by Achilleos et al., (2016) is currently the most exhaustive one and was based on 24-h integrated
($PM_{2.5}$ and $PM_{10}$) filter samples collected every 3 days for a period of one year (2012) in four cities in Cyprus. This study
concluded that Cypriot cities, like many others in Europe, are characterized by a major contribution of regional sulfate and
local (urban) emissions from traffic and domestic heating biomass burning.
Herewith, a detailed description of submicron (<1μm, $PM_1$) chemical composition and the further source apportionment of BC
and OA is presented for the first time in Cyprus. State-of-the-art on-line instrumentation (e.g., Q-ACSM, Aethalometer) was
deployed to investigate the temporal variability of aerosol composition at a location representative of the urban background
pollution in the capital city of Nicosia. Source apportionment of submicron organic aerosols was performed using the organic
fragments of the ACSM and Positive Matrix Factorization (PMF). The consistency of these results was assessed against the
chemical analysis of parallel filter samples and on-line measurements of external tracers. This study was extended to a 6-month
duration in order to cover the two main seasons of the semi-arid Eastern Mediterranean climate (short, mild and wet winter vs.
long, hot and dry summer), offering a comprehensive understanding of the daily and monthly variability of local and regional
sources of carbonaceous aerosols. Cold and warm periods were compared to highlight the complexity of local (combustion)
sources and the importance of regional ones. These results were further processed to apportion Black Carbon sources in Nicosia
with emphasis on local versus regional contribution.

## 92    2. Material and Methods

### 93    2.1 Sampling site

*Cyprus*: Cyprus is the third largest island in the Mediterranean Sea, extending approximately 240km long from east-to-west
and 100km wide. The closest countries and their distance from the capital city of Nicosia are respectively Turkey (110km),
Syria (250km), Lebanon (250km), Israel (300km), Egypt (400km), Jordan (430 km), and Greece (900 km from the Greek
mainland), (Fig. 1a).
The population of Cyprus (approximately 1 million inhabitants) is rather small compared to its neighbouring countries and the
rapidly growing (overall 400 million) population of the region (Lelieveld et al., 2013). The main urban areas of the island
shown in Fig. 1b, are those of Nicosia (c.a. 245,000 inhabitants), Limassol (c.a. 150,000 inhabitants), Larnaca (c.a. 50,000
inhabitants) and Paphos (c.a. 35,000 inhabitants). Cyprus has a Mediterranean and semi-arid climate with two main seasons:
a mild cold season (from December to March) and a hot warm season lasting about eight months (from April to November).
Rain occurs mainly in the cold season, with the warm one being extremely dry (i.e., almost no rain between May and
September)(Michaelides et al., 2018).
*Nicosia*: Nicosia is the largest city on the island and the southeasternmost of the European Union Member States' capitals.
Nicosia is currently partitioned in two, with a buffer zone in-between under the control of the United Nations; the southern
part being the capital of the Republic of Cyprus. The northern part of Nicosia (and the northern part of the island) is not
controlled by the government of the republic of Cyprus (Resolution 550, UN security council, 1984) (Fig. 1c). Geographically,
Nicosia is located in the centre of the island, within the Mesaoria plain, 150 m above sea level (asl), which is delimited on its
northern and southern edges by two mountain ranges; the Kyrenia Range culminating at 1,024 m asl, and the Troodos
Mountains culminating at 1,952 m asl, respectively. This topography channels winds within a more or less west-east corridor
(Fig. S5), feeding the city of Nicosia with long-range transported air masses from Europe, Africa, or the Middle East.
Measurements were performed at the Cyprus Atmospheric Observatory's Nicosia station (CAO-NIC) located at the Cyprus
Institute premises (Athalassa Campus; 174 m asl; 35.14N, 33.38E; Fig. 1c). The measurement site is considered an urban
background site, located within a low population density residential area with no significant local pollution hotspots in its
vicinity (i.e., no dense road traffic, industry, commercial centers, restaurants, etc.) and next to the Athalassa Forestry Park.
The period and duration of measurements presented here (07 December 2018 - 31 May 2019) were chosen to i) capture weather
conditions, atmospheric dynamics, and long-range pattern of the two main seasons, ii) investigate the contribution of domestic
heating emissions in winter, and iii) assess the potential increasing contribution of photochemical produced secondary aerosols
during the start of the dry and warm season. Local time (LT) in Cyprus is given as Eastern European Standard Time (EET)
(UTC+02:00 in winter and UTC+03:00 during the summer).

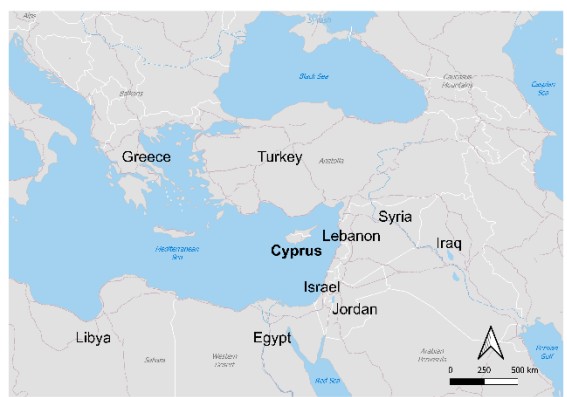

(a)

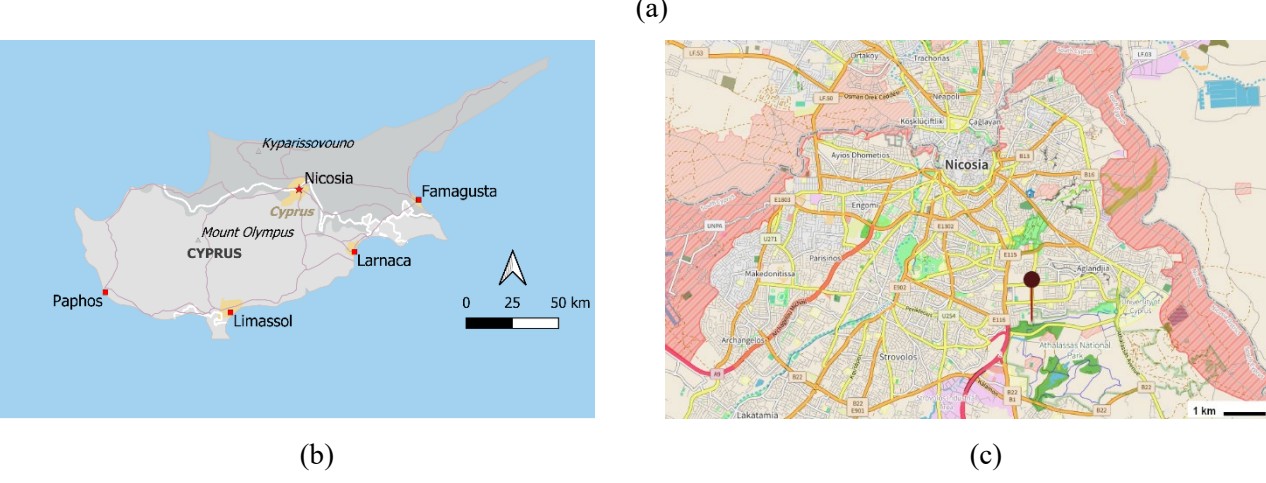

(b)                                                                                      (c)

**Figure 1: (a) Geographic location of the island of Cyprus and its closest Northern African and Middle Eastern neighbouring countries. (b) Location of the main cities of the Republic of Cyprus. Maps a,b were created by QGIS software v.3.26.3 utilizing the Natural Earth data (https://qgis.org). (c) Satellite view of the Nicosia agglomeration (grey area). The buffer zone dividing the island and the city is marked with red stripes; the location of the measurement site (CAO-NIC; The Cyprus Institute, Athalassa campus) is noted in red. (© OpenStreetMap contributors 2022. Distributed under the Open Data Commons Open Database License (ODbL) v1.0)**

## 2.2 On-line Aerosol Instrumentation

On-line aerosol instrumentation has been operated following the Standard Operating Procedures defined by the European Research Infrastructure on Aerosols, Clouds, and Trace Gases ACTRIS (https://www.actris.eu), and Cost COLOSSAL (CA16109, 2021).

Non-refractory submicron (NR-PM$_1$) aerosol chemical composition, i.e. organics, sulfate, nitrate, ammonium and chloride, was continuously monitored using a Quadrupole ACSM (Aerosol Chemical Speciation Monitor; Aerodyne Research Inc.) at a 30-min time resolution (Ng et al. 2011a). The instrument, along with a scanning mobility particle sizer (SMPS, described below), sampled through a sharp cut cyclone operated at 4 L min$^{-1}$ (SCC 1.197, BGI Inc., USA), and was equipped with a PM$_1$ aerodynamic lens, yielding an aerosol cut-off diameter of approximately 1.3μm. Data were retrieved using ACSM local v.1.6.0.3, implemented within Igor Pro (v. 6.37, Wavemetrics Inc., USA). The ACSM is designed and built around similar technology as the aerosol mass spectrometer (Jayne et al., 2000), where an aerodynamic particle focusing lens is combined with particle flash vaporization in high vacuum on the surface of a standard tungsten vaporizer heated at 600 °C, followed by electron impact ionization, separation and final detection of the resulting ions using a quadrupole mass spectrometer. Mass concentrations are corrected for incomplete detection due to particle bounce using the chemical composition-dependent collection efficiency (CDCE) (Middlebrook et al., 2012). The determined parameters, response factor (RF) and relative ionization efficiency (RIE) are reported in table S2.

Black carbon (BC) measurements were conducted using a 7-wavelength aethalometer (AE-33 Magee Scientific, US) at a 1-min time resolution. The aethalometer sampled ambient aerosol through a PM$_{2.5}$ aerosol inlet (SCC 1.829, BGI Inc., USA) at

a flow rate of 5 L min$^{-1}$ after passing through a nafion dryer. The instrument internally corrected the filter loading effect in
real-time, while a fixed value ($C_0$=1.39) was applied to compensate for the multi-scattering effect (Drinovec et al., 2015). BC
was apportioned to source specific components, namely $BC_{ff}$ related to fossil fuel combustion and $BC_{wb}$ related to wood
burning, by applying the "aethalometer model" (Sandradewi et al., 2008) on the 470 – 950 nm wavelength pair. The
instrument's default values for fossil fuel combustion and wood-burning aerosol Absorption Ångström Exponent, $AAE_{ff}$=1
and $AAE_{wb}$=2, respectively were selected after performing a sensitivity analysis on the AAE values (Supplement Section 3).

## 2.3 Ancillary measurements

**SMPS**: Particle number size distributions were monitored using a scanning mobility particle sizer (SMPS) consisting of an
electrostatic classifier (model 3080, TSI Inc., USA) coupled with a condensation particle counter (CPC; model 3070, TSI Inc.
USA) operating at a 5-min time resolution and at a 1 L min$^{-1}$ sample flow rate, measuring particles with a diameter ranging
from 9 to 700 nm. Ambient aerosols were drawn through a nafion dryer, and placed upstream, keeping sample RH below 30
%. Volume concentrations of assumed spherical particles derived by the SMPS were converted into mass concentrations using
a variable density calculated by the methodology described in Bougiatioti et al. (2014). The respective mass fractions time
series of chemical species were calculated based on the ACSM measurements. A density value of 1.77 g cm$^{-3}$ was used for
ammonium sulfate, and 1.35 g cm$^{-3}$ for organics (Florou et al., 2017; Lee et al., 2010), the two dominant compounds of $PM_1$
in Nicosia as detailed further below.
**Filter sampling:** Co-located 24h $PM_{2.5}$ samples were collected on quartz fiber filters (Tissuquartz, 47mm diameter, Pall) using
a low volume sampler (Leckel SEQ47/50) operating at a flowrate of 2.3 m$^3$ h$^{-1}$. The filter samples were analysed for i) organic
and elemental carbon using an OC/EC Lab Instrument (Sunset Laboratory Inc., OR, USA) implementing the EUSAAR II
protocol (Cavalli and Putaud, 2008), ii) carbohydrates, including levoglucosan, mannosan, galactosan, using an Ion
Chromatography Pulsed Amperometric Detection method (Thermo - Model ICS-3000) and iii) anions (Cl$^-$, NO$_3^-$, SO$_4^{2-}$, MSA,
Oxalate) and cations (K$^+$, Na$^+$, NH$_4^+$, Mg$^{2+}$, Ca$^{2+}$) using ion chromatography (Thermo - Model ICS-5000).
**Proton Transfer Reaction - Mass Spectrometry (PTR-MS):** Air was sampled through a 20m long, 3/8" o.d. (1/4" i.d.)
sheathed Teflon line that ran from the roof of the building to the instrument. A Teflon filter (0.2μm diameter porosity) was
installed at the inlet to prevent large aerosol particles and insects from entering the sampling line. The resulting residence time
of air in the line was estimated to be approximately 0.5 min. The temporal resolution of Volatile Organic Compounds (VOCs)
measured by the PTR-MS (Ionicon Analytik, Austria) was approximately two minutes (the time required to measure 55
different ions at 2 seconds per ion). The basic operation principles of the PTR-MS instrument have been described in detail by
Lindinger et al. (2011). Briefly, a stable flow of air and high concentrations of H$_3$O$^+$ ions are continuously sampled into a drift
tube held at 2.2 mbar pressure. There, compounds with a proton affinity greater than water, including a large selection of
Oxygenated Volatile Organic Compounds (OVOCs), undergo efficient proton-transfer reactions with the H$_3$O$^+$ ions to produce
protonated organic product ions, which can be detected by a mass spectrometer.
**Meteorological Parameters:** Standard meteorological parameters (temperature, relative humidity, wind speed and direction)
were obtained from the meteorological station of the Cyprus Department of Meteorology, installed 10 m above ground, located
at the Athalassa Forestry Park (164 m asl) lying approximately 1.3 km east of the CAO-NIC station. Wind speed and direction
data were further used in this study for component-specific non-parametric wind regression analysis (NWR) performed using
the ZeFir toolbox (Petit et al., 2017) developed within the Igor Pro software (Wavemetrics Inc.). A co-located automatic
CIMEL CE370 micro-LIDAR was operated continuously to retrieve the Planetary Boundary Layer Height (PBLH) and better
assess the influence of atmospheric dynamics on in-situ ground-based observations.
**Air masses back trajectory analysis:** Five-day air mass back trajectories arriving at 1000m altitude above the sampling site
were computed every 6 hours, using the Hybrid Single-Particle Lagrangian Integrated Trajectory model (HYPLIT4; Stein et
al., 2015) using the Global Data Assimilation System (GDAS 1) meteorological data fields (with 1° spatial resolution). Back
trajectories were coupled to measured concentrations, assessing origins and source contributions to specific chemical
components, by applying the Potential Source Contribution Function (PSCF) technique as implemented in the ZeFir toolbox
described above.
**2.4. Source Apportionment analysis**
Positive Matrix Factorization (PMF) is an advanced multivariate factor analysis tool that attempts to identify the contributing
factors, or sources, of atmospheric pollutants at a sampling site. For this study, source apportionment was performed on the
organic mass spectra dataset collected by the ACSM. The (PMF) method (Paatero and Tapper, 1994) using the multilinear
engine (ME-2) model developed by Paatero (1999) was implemented using the SoFi (Source Finder) toolkit (SoFi 6D;
Canonaco et al., 2013). PMF allows the decomposition of the OA mass spectra matrix X into two matrices, G and F and a
remaining residual matrix, E:
$X = G * F + E$    (1)
Where X is the input dataset matrix (measured quantity), F is the resulting source profile matrix, G is the source contribution
matrix (temporal variability of each source), and E represents the model residual matrix. Based on a number of criteria, the
optimal solution is selected, aiming at being physically meaningful that can be supported by external indicators (ancillary
measurements), and trying to minimize values in the residual matrix E. Model input data and error matrices (in μg m$^{-3}$), were
exported using the ACSM software. Data points with a signal-to-noise (S/N) ratio smaller than 0.2 were removed. Points with
S/N between 0.2 and 2 were down-weighted by increasing their estimated error values (Ulbrich et al., 2009; Paatero and Hopke,
2003). M/z (mass-to-charge ratio) values ranging from 10 to 120 were used in the analysis. $CO_2$-related variables were
excluded from the PMF and finally reinserted into the solution.
Source apportionment of OA was performed following the general steps described by Crippa et al. (2014) and the recently
updated harmonised standard operating procedures for seasonal OA PMF (Chen et al., 2022). As a first step, unconstrained
PMF analyses were performed with a number of factors ranging from 2 to 8 in order to identify the most relevant number of
factors and potential sources.  If primary organic aerosol factor profiles such as Hydrocarbon-like OA (HOA) or biomass
burning-like OA (BBOA) were found, then the corresponding site-specific primary OA (POA) mass spectra (see discussion
below) or spectra found in the literature (e.g., Ng et al., 2011 and Crippa et al., 2014) were set as constrains in the PMF, using
the "a-value" approach (Paatero and Hopke, 2009; Canonaco et al., 2013). A sensitivity analysis was then performed with
different a-values to assess the level of constrain introduced in each factor with i) a constrained HOA using, as an anchor the
HOA spectrum found in Ng et al. (2011) with the a-values ranging between 0.05 and 2.0, ii) a constrained BBOA factor with
the a-values from 0.2 to 0.5 from Ng et al. (2011), and iii) a constrained cooking OA (COA) factor from Crippa et al. (2014)
with a-values from 0.2 to 0.5. Once this sensitivity analysis was completed, the evaluation of the PMF results showed that the
BBOA factor could not account for the entire m/z 60 mass fragment, which fragment was distributed within 2 factors.
Additionally, the correlation of BBOA with $BC_{wb}$ showed to be unsatisfactory (section S1). On the other hand, given the
BBOA factor's sensitivity to the type of solid fuel used, different biomass-burning factor profiles have been reported in various
regions around the world (Xu et al., 2020; Trubetskaya et al., 2021). Consequently, a site-specific BBOA factor profile
($BBOA_{cy}$) was selected. The $BBOA_{cy}$ spectrum was calculated as an average of  20 PMF runs from the initial unconstrained
PMF for the cold period, validated by it's time-series correlation to $BC_{wb}$. Since aged OA (i.e. Oxygen-like OA, OOA) factors
show more variability between measurement sites in terms of their mass spectra, no constrain was introduced for these factors
(Canonaco et al., 2015).
In this study, the BBOA factor - a major contributor of OA during winter - could not be properly resolved when performing
the PMF analysis on the entire period dataset. A seasonal approach was followed instead, separating the OA dataset into two
periods that were then used to describe both the two periods (cold and warm, respectively). The criteria used to delineate those
two periods are presented and discussed in section 3.2.
One factor was consequentially constrained with the resulting BBOA$_{cy}$ spectrum (with an a-value in the 0-0.5 range, using
steps of 0.02), obtaining the optimal solution using an a-value equal to 0.46. A widely referred-to standard mass spectrum (Sun
et al., 2016; Duan et al., 2020) derived from Ng et al. (2011) was used to constrain the HOA factor, with an a-value of 0.2,
thus obtaining the best correlation with BC$_{ff}$, a tracer for traffic-related emissions. A detailed description of the OA source
apportionment analysis can be found in section S1 in the supplementary material.
**3. Results and Discussion**
**3.1. On-line aerosol data quality check**
A chemical mass closure exercise for PM$_1$ was performed at a temporal resolution of 1h to check the quality of the on-line
aerosol measurements. Chemically reconstructed PM$_1$ was calculated as the sum of the mass concentration of all non-refractory
species measured by the ACSM (OA, NO$_3^-$, SO$_4^{2-}$, NH$_4^+$, Cl$^-$) plus the BC concentrations measured by the Aethalometer AE-
33 (Putaud et al., 2004). The contribution of other chemical constituents to submicron aerosols, such as sea salt and dust
(measured by co-located filter sampling), was found to be low and therefore neglected here. A scatter plot of the ACSM + AE-
33 measurements vs. the SMPS-derived PM$_1$ concentrations is shown in Figure S4a. The results indicate a very good
correlation (r$^2$ = 0.88; N=1823) and a slope of 1.2 (Fig. S4a). This 20% discrepancy lies within the uncertainty of the on-line
instruments. It could be attributed to the cut-off size of the SMPS at 700nm, which is slightly lower compared than the ACSM.
In addition, ACSM individual chemical species were compared with co-located off-line analyses performed on daily PM$_{2.5}$
filters. As shown in Fig. S4b-e, very good agreement was obtained between on-line and off-line measurements with r$^2 \geq 0.80$
(N=165-175) for all species. The discrepancy between ACSM and filter measurements for nitrate (slope of 1.3) could be
attributed to the volatilization of HNO$_3$ from the filter surface due to the presence of semi-volatile ammonium nitrate. The
obtained slopes for ammonium and sulfate below 1:1 (0.81 and 0.85, respectively) are consistent with the fact that fine
(NH$_4$)$_2$SO$_4$ aerosols, mainly originating from secondary processes and long-range transport (Sciare et al., 2010; Freutel et al.,
2013), can be found at a large size mode possibly exceeding 1 μm, consequently not being sampled by the ACSM.
The study investigated the aerosol ion balance using both online and offline inorganic measurements. The ratio of the measured
concentration of NH$_4^+$$_{Measured}$ and the estimated concentration of NH$_4^+$$_{Predicted}$, as calculated in Jiang et al., (2019), was used for
this purpose. The results showed a slope of 0.80 for online measurements and 0.96 for offline measurements. These findings,
suggest that the atmospheric aerosol observed during the study period was predominantly neutral, taking into account the
uncertainties of ammonium concentrations reported in Q-ACSM intercomparison studies (Crenn et al., 2015), as well as the
species' relatively high detection limit (Ng et al., 2011).
An interesting result obtained from the comparison of OA (ACSM) with OC (from filters) is an OM-to-OC ratio of 1.42 which
is at the lower end of ratios reported for urban environments, which usually exhibit typical values of 1.6 ± 0.2 (Petit et al.,
2015; Theodosi et al., 2011; Brown et al., 2013). Without neglecting the fact that two different size fractions were compared
(PM$_1$ for the ACSM and PM$_{2.5}$ for the filter sampling), this low ratio probably point to long-chain hydrocarbon OA that often
are related to primary combustion (poorly oxidized) OA (Aiken et al., 2008). As such, this ratio could represent an independent
means of verification of the consistency of our source apportionment between primary and secondary OA.
Finally, black carbon concentrations derived from light absorption measurements (Aethalometer AE-33) were compared
against filter-based EC measurements (see Fig. S4f). Data from the two techniques correlate very well (r²=0.83), with a BC/EC
ratio of 1.67 being similar to studies in other urban areas (Rigler et al., 2020; Liu et al., 2022), highlighting the existence of a
BC absorption enhancement (E$_{abs}$) attributable to a lensing effect induced by other chemical species, among which secondary
OA may play an important role (Zhang et al., 2018).

## 3.2 Meteorological conditions

**Delineation of cold vs. warm seasons**: The ACSM organic mass at m/z 60 is characteristic of the fragmentation of levoglucosan, a product of cellulose pyrolysis and well-established biomass burning marker (Alfarra et al., 2007). Its respective contribution to total OA ($f_{60}$) was used in this study as an indicator of biomass burning for domestic heating to delineate cold vs. warm seasons, comparing with the 0.3% threshold proposed by Cubison et al. (2011) for air masses influenced by biomass burning. Except for a single small peak in early May, corresponding to open fires for the celebration of the Greek Orthodox Easter, the last instance when $f_{60}$ was above the threshold was recorded during the first week of April (Fig. 2). From then onwards, daily air temperature started rising constantly, from about 15°C at the beginning of April up to to 30°C at the end of May. These two features dictated the division of the dataset into two periods: a cold period of four months (07/12/2018-08/04/2019), with an average temperature of 12 ± 4ºC, and a warm period of two months (09/04/2019 – 31/05/2019), with an average temperature of 20 ± 7ºC.

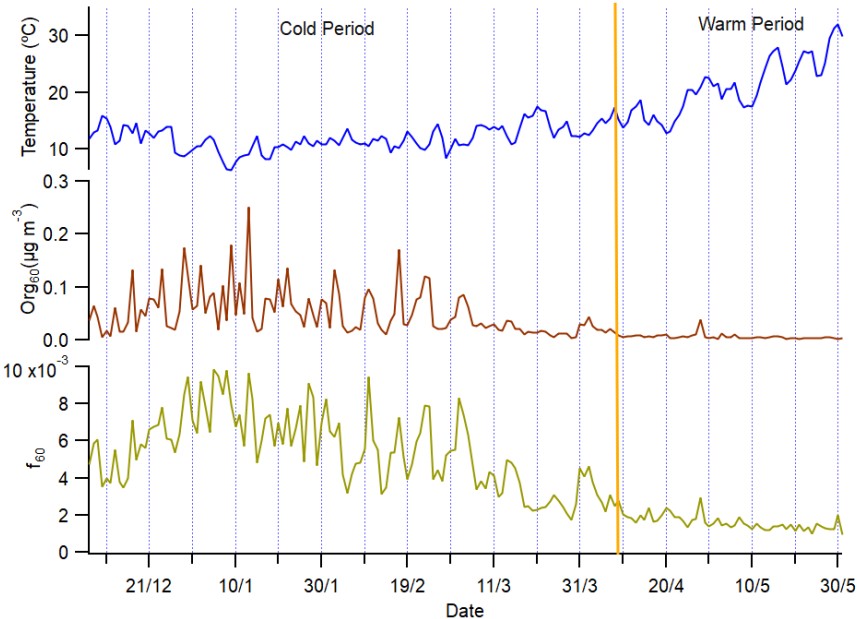

**Figure 2:** Time series of air temperature (blue), m/z 60 organic concentration (org60, brown) and $f_{60}$ fragment (green) for the cold and warm periods. The vertical line is used to delineate the measurements within the two seasons.

**Wind sectors**: During these two periods, a distinct pattern in the wind sectors and the air masses arriving at the sampling site was observed. As seen in Fig. S5, the dominant wind direction for the cold period was the NW-SW [225 ° - 315 °] sector encompassing 48% of the total wind directions, while the NE-SE [45 ° - 135 °] sector covered 26%. During the warm period, the weight of this proportion is shifting even more towards the NW-SW [225 ° - 315 °] sector, having a 62% of total air masses while only 17% are arriving from the NE-SE [45 ° - 135 °] sector.

**Air mass origin**: A cluster analysis was performed (Fig. S6a,b) for both periods in order to better assess the main upwind regions responsible for long-range transported air pollution over Cyprus and their change relative to the period of the year. The number of clusters used in each season was determined by considering the percentage change in Total Spatial Variance (TSV) as a function of the number of clusters of merged trajectories (Fig. S6c,d) and the mean trajectory paths of each cluster (Fig. S6e,f). The first large drops observed in TSV from the two – to – three and the three – to – four cluster transition could not represent all the recorded trajectories and especially the ones describing air masses arriving in Nicosia from the east. The next remarkable decrease in TSV was recorded when moving to seven clusters. Thus, for both periods, seven clusters were chosen to better represent all air masses arriving in Nicosia.

A significant part of most of the calculated mean trajectory path representing clusters arriving in Cyprus was found to be related to the wider western sector, with many of them though, passing over Turkey before reaching Cyprus. Interestingly, this analysis showed one cluster (Cluster 1) arriving from the Middle East (close to Lebanon and Syria) and another four (Cluster

1, 2, 5, 6) passing over the western part of Turkey for the cold period. For the warm period, the only clusters arriving from the
Middle East were the ones  related to Turkey (Clusters 1, 5, 6). Plotting all individual 72h back trajectories (Fig S6e,f) showed
that a clear portion (almost 25% of the calculated trajectories) are being influenced by the Middle East, especially for the cold
period (Fig S6e).
**3.3. Chemical composition of PM₁**
**Seasonal perspective of PM₁**: The time series of PM₁ chemical composition derived from the ACSM (OA, $SO_4^{2-}$, $NH_4^+$, $NO_3^-$
, $Cl^-$) and the Aethalometer ($BC_{ff}$, $BC_{wb}$) are depicted for the entire measuring period in Figure 3. Averaged data  (6h averaging
period) are shown here for clarity. Furthermore, the relative average contribution of each chemical constituent to total PM₁
concentrations is depicted in the respective inner pie charts for both periods.

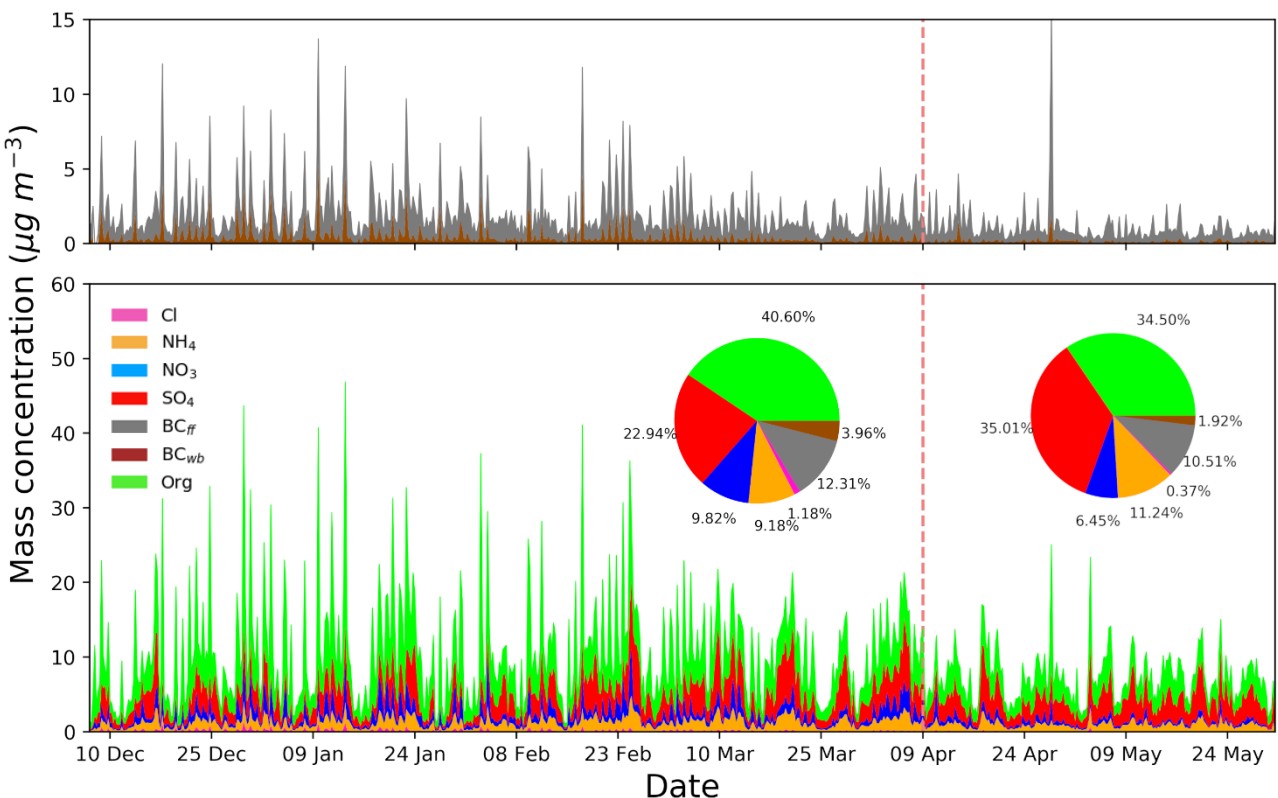


**Figure 3: Stacked area plots of the chemical composition time series for PM₁ in Nicosia derived from 6-hour averages of ACSM and**
**AE-33 measurements. The vertical dashed red line separates the cold from the warm season. The average relative contribution of**
**each species is shown in the respective pie charts (inner panels) for each season.**
Although intense and short-duration peaks are observed for carbonaceous aerosols (OA, $BC_{ff}$, $BC_{wb}$), background NR-PM₁
concentration levels (between peak values) remain well below 10 µg m⁻³ for the 6-h average in both seasons. In other words,
no PM₁ pollution episodes (with e.g., concentrations above 10 µg m⁻³) lasting for consecutive days were observed. Such lack
of intense and persistent PM₁ pollution episodes differs from what is reported in central and northern Europe, where stagnant
(anticyclonic) conditions occur together with continental (polluted) air masses, mainly in winter and springtime (e.g., Petit et
al., 2015). This suggests that the relatively low emissions from Cyprus (compared to the neighboring countries) and its remote
marine location (i.e., far from densely populated areas) may prevent the build-up of high PM₁ pollution events over Nicosia.
On the other hand, clear differences can be observed between both periods, with significantly higher PM₁ concentrations during
the cold period, associated with repeated, intense peaks of OA and BC - not observed during the warm season – and suggesting
local combustion emissions. The highest PM₁ concentrations were observed between December 28[th] 2018 and January 13[th]
2019 (Fig. 3) and were associated with low temperatures and Christmas holidays, both likely to promote domestic heating use.
During the warm period, the higher contribution of sulfate, and lower contribution of OA, are clearly noticeable. The

contribution of nitrate during the warm period, most probably in the form of semi-volatile $NH_4NO_3$, remains marginal, possibly due to non-favourable thermodynamic conditions preventing its formation and accumulation.

**PM$_1$ chemical composition**: For the cold period, the average calculated mass concentration of $PM_1$ (calculated as the sum of chemical components measured by AE-33 and ACSM) was $12.35 \pm 9.77$ µg m$^{-3}$, with $10.34 \pm 7.92$ µg m$^{-3}$ being the average concentration of the non-refractory species (Table 1). OA constitutes the larger fraction of $PM_1$ mass, with an average concentration of $5.03 \pm 5.48$ µg m$^{-3}$ (41 %), followed by sulfate (23 %), black carbon (16 %), nitrate (10 %), ammonium (9 %), and chloride (1 %). These concentrations and the overall distribution of chemical components in NR-$PM_1$ are similar to those measured by ACSM in other European cities (Bressi et al., 2021). Concentrations appear to decline during the warm period, with an average calculated $PM_1$ concentration of $8.18 \pm 4.65$ µg m$^{-3}$, including $7.18 \pm 3.81$ µg m$^{-3}$ from the non-refractory components. The dominant species during the warm period were sulfate and OA, each representing 35 % of $PM_1$, followed by black carbon (12 %), ammonium (11 %) and nitrate (6%). During that period, chloride concentrations were negligible, contributing less than 1 % (Table 1).

**Table 1: Species mean, standard deviation, median concentrations and respective contribution to PM$_1$ during cold and warm periods in Nicosia.**

| µg m$^{-3}$ | Cold Period | | | | Warm Period | | | |
|---|---|---|---|---|---|---|---|---|
| | Mean | Std | Median | Contribution (%) | Mean | Std | Median | Contribution (%) |
| OA | 5.03 | 5.48 | 3.35 | 41 | 2.83 | 1.91 | 2.51 | 35 |
| $SO_4^{2-}$ | 2.84 | 1.89 | 2.60 | 23 | 2.87 | 1.50 | 2.61 | 35 |
| $NO_3^-$ | 1.22 | 1.25 | 0.75 | 10 | 0.53 | 0.56 | 0.34 | 6 |
| $NH_4^+$ | 1.14 | 0.77 | 1.01 | 9 | 0.92 | 0.55 | 0.84 | 11 |
| $Cl^-$ | 0.14 | 0.21 | 0.07 | 1 | 0.03 | 0.08 | 0.01 | <1 |
| BC | 2.01 | 2.31 | 1.26 | 16 | 1.01 | 1.46 | 0.66 | 12 |
| $PM_1$ | 12.35 | 9.77 | 10.01 | 100 | 8.18 | 4.65 | 7.53 | 100 |

Interestingly, sulfate concentrations recorded in Nicosia are higher compared to what is commonly observed in other European countries and Mediterranean cities (Table 2) and likely reflect a regional pattern of sulfur-rich emissions compared to Europe, where $SO_2$ emissions have strongly decreased during the last decades (Smith et al., 2011; Chin et al., 2014) thanks to the implementation of specific abatement measures on reducing sulfur emissions (European NEC Directive (EU, 2016) and United Nation Gothenburg (1999) protocol). More specifically, the importance of sulfur emissions in Turkey (2 455 Gg, EEA 2021), which were 50% higher compared to the total $SO_x$ emissions of the EU 28 in 2019, together with the fact that half of the air masses reaching Cyprus are passing over Turkey (see Fig. S6) are key contributors to the high concentrations of sulfate in our study.

Shipping emissions appear to have a relatively minor impact on the concentration of sulfate. To more accurately determine the contribution of shipping emissions to $SO_4^{-2}$, $SO_2$, and total $PM_{2.5}$ a supplementary analysis was conducted using the WRF-Chem model, which simulates both physical and chemical processes occurring in the atmosphere. This model has been extensively evaluated in several studies for the Eastern Mediterranean (Kushta et al., 2018) and Europe (Berger et al., 2016; Tuccella et al., 2012). Following the set-up used in Giannakis et al., (2019) and driven by the EDGAR v.5 anthropogenic emission inventories (Crippa et al., 2019), two annual-long simulations were performed: firstly, including all sectoral emissions in the model (baseline simulation So) and a second simulation where shipping emissions have been omitted (scenario simulation, $S_1$) to identify the impact of shipping on gaseous and aerosol sulfur-related species concentrations ($SO_2$ and $SO_4^{-2}$) and total $PM_{2.5}$ over the Central and Eastern Mediterranean. The figures (S final) describe the contribution of shipping in absolute terms (Fig. S7 a,c,e) and as a percentage (Fig. S7 b,d,f) for the $SO_4^{-2}$, $SO_2$, and total $PM_{2.5}$ calculated for each species. According to these results, the highest impact of shipping on near-ground modelled concentrations of the three species ($SO_4^{-2}$, $SO_2$ and $PM_{2.5}$) was estimated along the central Mediterranean region (yellow grids, west of the Balkans and Greece), as

well as a small section south of Greece. The Levantine basin, where Cyprus is located, experiences significantly lower influence under the no-shipping emissions sensitivity test. More specifically, over the East Mediterranean, $SO_4^{-2}$ concentrations represent a relative change of only about 6-8% when including shipping emissions.

**Table 2: Comparison of concentration, and percentage contribution to PM$_1$, between the main submicron chemical species derived by ACSM.**

| | PM$_1$ ($\mu g\ m^{-3}$) | OA ($\mu g\ m^{-3}$) | SO$_4^{2-}$ ($\mu g\ m^{-3}$) | NH$_4^+$ ($\mu g\ m^{-3}$) | NO$_3^-$ ($\mu g\ m^{-3}$) | Cl$^-$ ($\mu g\ m^{-3}$) | Reference |
|---|---|---|---|---|---|---|---|
| Nicosia Cold (DJFM) | 12.32 | 5.03 | 2.81 | 1.14 | 1.22 | 0.31 | This study |
| Nicosia Warm (AM) | 8.18 | 2.83 | 2.87 | 0.92 | 0.53 | 0.05 | This study |
| Cyprus RB* (Annual) | 7.6 | 3.26 | 2.66 | 0.98 | 0.23 | - | Chen et al. (2022) |
| European UB** (Annual) | 10.6 | 5.3 | 2.0 | - | 1.9 | - | Bressi et al. (2021) |
| S. Europe RB*** (Annual) | 6.3 | 3.5 | 1.3 | - | 0.8 | - | Bressi et al. (2021) |
| Athens Winter | 18.7 | 13.13 | 2.4 | - | 1.8 | 0.14 | Stavroulas et al. (2019) |
| Athens Spring | 6.42 | 3.3 | 2.1 | 0.6 | 0.4 | 0.02 | Stavroulas et al. (2019) |
| Marseille Winter | 11.9 | 6.17 | 1.12 | 0.86 | 1.58 | 0.09 | Chazeau et al. (2021) |
| Marseille Spring | 8.09 | 3.86 | 1.06 | 0.70 | 1.13 | 0.04 | Chazeau et al. (2021) |
| Barcelona (Annual) | 9.85 | 4.10 | 1.70 | 1.05 | 1.35 | 0.06 | Via et al. (2021) |

* Cyprus Regional background

** European urban background = Barcelona (Spain) + London (UK) + Prague (Czech) + Tartu (Estonia) + Zurich (Switzerland)

***Southern European regional background = Ersa (Corsica, France) + Finokalia (Crete, Greece)

The main difference between the cold and warm periods lies in the decrease in the concentration of carbonaceous aerosols (OA, BC) and $NO_3^-$ by almost a factor of two. Several phenomena can explain this significant seasonal variation: the absence of a domestic heating source (mainly biomass burning as explained in Fig. 2); the absence of Middle East air masses during the warm period (see discussion later on); the increase in the Planetary Boundary Layer Height (PBLH) above Nicosia (Fig. S8) enhancing vertical dilution of local emissions during the warm period and therefore lowering ground-based concentrations; less favourable thermodynamic conditions, with warmer and dryer air, also preventing the condensation of semi-volatile species (e.g., ammonium nitrate). Sulfate concentrations do not exhibit a similar seasonal pattern and therefore seem to be less affected by the above factors. On the contrary, the increase in photochemistry enhances the formation of sulfate aerosols, and the decrease in precipitation enhances aerosol lifetime, strengthening the impact of long-range transport.

### 3.4. Diurnal variability of PM$_1$ chemical constituents

Figure 4 shows the diurnal variability of the PM$_1$ species derived from the ACSM and AE-33 for both the cold (Fig. 4a) and warm (Fig. 4b) periods. The diurnal variability of the apportioned BC related to fossil fuel combustion (BC$_{ff}$) and wood-burning (BC$_{wb}$) are also depicted here.

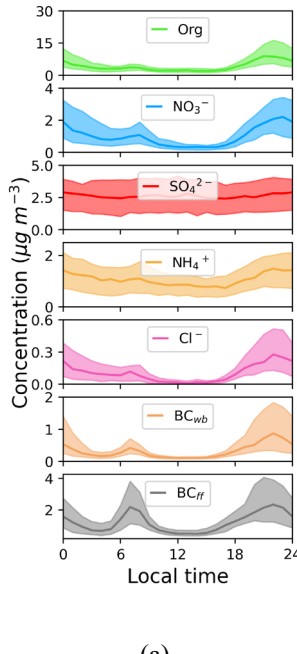 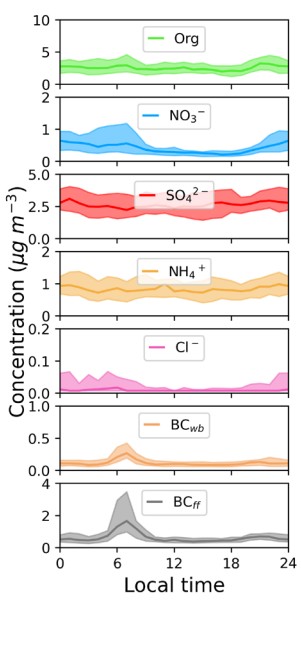

(a)                                         (b)

**Figure 4:** Median diurnal trends of the main submicron chemical constituents (OA, $SO_4^{2-}$, $NO_3^-$, $NH_4^+$, $Cl^-$ and BC) during the a) cold and b) warm periods. The shaded area represents the 25th and 75th percentiles of the diurnals.

**Organic aerosols**: Organic aerosols dominate the cold period $PM_1$ concentration levels, exhibiting a night-time maximum above 12 µg m$^{-3}$ and a second smaller maximum at 4 µg m$^{-3}$, coinciding with local traffic rush hour (06:00-09:00 LT). Elevated OA concentrations in the cold period during the night (max at 22:00 LT) are a common, well-documented feature in many urban environments across Europe and the Mediterranean (e.g., Florou et al., 2017; Stavroulas et al., 2019; Chazeau et al., 2021). They can be attributed to higher emissions from domestic heating, evening traffic peak and cooking activities. The strong correlation between OA and $BC_{wb}$ ($R^2$=0.81; N=2934; Fig. S9) suggests that residential wood burning is an important contributor to this nighttime peak. Interestingly, this peak is not significantly amplified by a lower PBLH during night-time, which seems to remain relatively stable with no significant diurnal variability during the cold period (Fig. S8). It is also worth noting that background OA concentrations observed both at the end of the night and middle of the day, when local emissions are minimal, remain relatively high at around 3 µg m$^{-3}$. The diurnal variability of OA is much less pronounced during the warm period, suggesting a more important contribution of regional sources to OA compared to the strong dynamic of local emissions. The assumption of a more important contribution from regional OA during the warm period is further supported by a mean OA concentration of 2.83 µg m$^{-3}$ (Table 2) that is close to the averaged OA concentrations of 3.2 µg m$^{-3}$ reported for a 2-year period continuous observations with Q-ACSM (2015-2016) at the rural background site of the Cyprus Atmospheric Observatory at Agia Marian Xyliatou (CAO-AMX), at a distance roughly 40 km from Nicosia (Chen et al., 2022). During the warm period, a small OA peak remains visible in the morning, with a similar amplitude to the cold season, likely to be related to traffic emissions. A second peak can be observed at 21:00 LT (not observed in BC), potentially originating from cooking activities. Heavy oil combustion from shipping could possibly contribute to OA. Further to the poor contribution of shipping emission on OA, a model study of sources of organic aerosols in Europe using CAMx (Jiang et al., 2019) showed that the contribution of "other anthropogenic sources" (gathering shipping, industry, and energy production) on OA (POA+SOA) was, typically, of the order of 10% during summer and winter in the Eastern Mediterranean region close to Cyprus. Based on a simple receptor model, $PM_{2.5}$ source apportionment performed in Nicosia, Achilleos et al. (2016) showed that the contribution from shipping is approximately 8% to $PM_{2.5}$. Most of the transported mass is attributed to $SO_4^{2-}$ with a minor contribution from carbonaceous aerosols. In conclusion, shipping emissions are likely to play a minor role in OA concentrations.

**Black carbon**: During the cold season, BC follows a bimodal diurnal pattern, which can be further apportioned by focusing on its source-specific components $BC_{ff}$ and $BC_{wb}$. The fossil fuel component exhibits two maxima, one in the early morning, coinciding with traffic rush hour, and one in the late afternoon, most probably related to both traffic and an increase in energy demand due to domestic heating (see discussion later on). On the other hand, $BC_{wb}$ diurnal variability is dominated by a night-time maximum (20:00 - 01:00 LT), peaking one hour after $BC_{ff}$ and linked to wintertime residential wood-burning emissions, contributing up to 33 % of total BC. During the warm season, the BC diurnal pattern is characterised by the absence of a night-time maximum while still exhibiting a significant peak in the morning, dominated by $BC_{ff}$. The very low contribution of biomass-related combustion particles during the warm period, as previously noted from m/z 60 in Fig. 2, is further supported here, with $BC_{wb}$ exhibiting a nearly flat diurnal variability with close-to-zero mass concentrations. The contribution of shipping in the Mediterranean on Black Carbon (BC) concentrations was investigated from model estimates by Marmer et al. (2009) based on three (3) most commonly used ship emissions inventories: 1) EDGAR FT by Olivier et al. (2005), 2) Eyring et al. (2005), and 3) EMEP by Vestreng et al. (2007). Results showed that shipping emissions were contributing to typically 15-25% of BC in the E. Mediterranean, far from the shipping routes (which is the case for Cyprus). A similar result was found from a more detailed (Positive Matrix Factorization) $PM_{2.5}$ source apportionment analysis performed in Nicosia in 2018, with heavy oil combustion contributing 7% to $PM_{2.5}$ (Bimenyimana et al., 2023 under review), and the relevant factor containing less than $0.1 \mu g \ m^{-3}$ of EC.

**Secondary inorganic aerosols**: During the cold season, non-refractory nitrate and chloride detected by the Q-ACSM are mostly present in the form of semi-volatile $NH_4NO_3$ and $NH_4Cl$ (Guo et al., 2017; Theodosi et al., 2018). They show a night-time maximum (Fig. 4-a), reflecting the presence of gas precursors ($NH_3$, $HNO_3$, $HCl$) and the more favourable thermodynamic conditions with lower temperatures, higher relative humidity, and condensation sink due to high PM concentrations of combustion aerosols (traffic, domestic heating). Additionally, there is a smaller morning $NO_3^-$ peak, most probably linked to traffic (Foret et al., 2022). This is not observed for chloride, suggesting that HCl may not be as abundant in the morning compared to the evening. The less favourable thermodynamic conditions during the warm period lead to very small concentrations of semi-volatile $NO3^-$ and $Cl^-$ (Fig. 4b). As expected, sulfate does not show a pronounced diurnal pattern, irrespective of the period, and pointing to regionally-processed aerosols (Fig. 4a,b).

### 3.5. OA Source Apportionment

### 3.5.1 OA source apportionment during the cold period

For the cold period, the optimal PMF result has been found using a 5-factor solution following the approach detailed in section 2.4. The identification of OA sources related to these 5 factors was then performed following the typical combination of information from i) OA mass spectra (Fig. 5a), ii) the correlation of each factor with source-specific tracers (see Fig. 5b), iii) their diurnal variability (Fig. 6a), and iv) their daily (week days vs. week-end) pattern (also Fig. 6b). The five factors were then assigned to the following sources: A primary BBOA (Biomass Burning Organic Aerosol), two primary HOA (Hydrocarbon-like Organic Aerosol; HOA-1 and HOA-2) and two secondary OA sources, namely low-volatile MO-OOA (More-Oxidized Oxygenated Organic Aerosol) and semi-volatile LO-OOA (Less-Oxidized Oxygenated Organic Aerosol). This source apportionment is presented and justified below for each factor:

**HOA-1 (Hydrocarbon-Like OA Type 1):** The mass spectrum of HOA-1 (Fig. 5a) is consistent with a fossil fuel (traffic) combustion source that can be identified by the prevailing contributions of the ion series representing $C_nH_{2n-1}$ (m/z = 27, 41, 55, 69, 83, 97, typical fragments of cycloalkanes or unsaturated hydrocarbon chains) and $C_nH_{2n+1}$ (m/z = 29, 43, 57, 71, 85, 99, typical fragments of alkane chains). Hence, this factor mass spectrum is well correlated to eight selected HOA factors related to vehicular traffic found in the literature (Fig. S10a) and relevant to European and Mediterranean environments. The traffic-related origin of the HOA-1 factor can be further confirmed by the good correlation with $BC_{ff}$ ($R^2$=0.65; N=2934; Fig.

S11a), benzene ($R^2$=0.72; N=1165; Fig. S11b). The diurnal variability of HOA-1 shows a bimodal cycle with a sharp maximum during the morning rush hour with an amplitude similar to $BC_{ff}$ (Fig. 6a), and a broader maximum in the evening possibly encompassing emissions from traffic and diesel-fired residential heating systems. In the weekly cycle, as depicted in Fig. 6b, the morning peak decreases on Saturday. It is nearly absent on Sunday mornings, aligned with the de-escalation of traffic emissions usually observed during weekend mornings.

**BBOA (Biomass Burning OA):** The mass spectrum of the site-specific BBOA factor (reported as $BBOA_{cy}$ in section 2.4) exhibits characteristic peaks at m/z 29, 60, and 73 (Fig. 5a), which are indicative of biomass burning (Crippa et al., 2014). The mass spectrum is quite similar to other BBOA spectra found in the Mediterranean and Europe (Fig. S10c), with a key difference here being the rather low contribution of a signal at m/z=43. The biomass burning-related origin of the factor is further confirmed by the strong correlation with $BC_{wb}$ ($R^2$=0.81; N=2934; Fig. S11c), benzene ($R^2$=0.61; N=1162; Fig. S11d) and levoglucosan ($R^2$=0.94; N=125; Fig, S11e) a typical tracer of biomass burning (Fourtziou et al., 2017). The BBOA diurnal pattern exhibits an expected well-marked night-time maximum around 22:00 LT, consistent with residential wood-burning activities. This night-time maximum is observed throughout the week (Fig. 6a), confirming the important role of wood burning for heating in the city. Interestingly, the higher concentrations of BBOA as well as $BC_{wb}$ (Fig. 6b) were observed on Sunday evenings, pointing to the recreational use of fireplaces, leading to enhanced residential wood-burning emissions during the weekend, a feature also reported in other sites in Europe and the US (Bressi et al., 2016; Rattanavaraha et al., 2017; Zhang et al., 2019).

**HOA-2 (Hydrocarbon-Like OA Type 2):** The mass spectrum obtained for this factor (Fig. 5a) is similar to the HOA-1 factor, with high signals for the ion series $C_nH_{2n+1}^+$ and $C_nH_{2n-1}^+$. The main differences between these two factors occur in the relative contribution of m/z 41 compared to m/z 43 and the relative contribution of m/z 55 compared to m/z 57, which are both much higher for HOA-2 than for HOA-1. Furthermore, the contribution of signal to m/z 44 is more significant in HOA-2, which can imply a mix of various sources and/or a possibly higher degree of atmospheric processing. Other discrepancies with HOA-1 concern its diurnal variability, with an intense maximum at night (Fig. 6a), and its average concentration levels, which are almost three times higher than HOA-1.

Influence of cooking activities: The HOA-2 diurnal profile has a small peak at 13:00 LT and a significantly higher one at 21:00 LT, effectively coinciding with typical meal times in Cyprus as well as those reported in the literature for Greece (Siouti et al., 2021), therefore indicating the influence of cooking activities to this factor. When plotting $f_{55}$ vs $f_{57}$ (Mohr et al., 2012) and colouring the data points by the corresponding time of day, a distinct pattern appears with data of higher $f_{55}$ over $f_{57}$ being clustered to the top left of the triangle, close to the fitted lines representing cooking (Fig. S12) and coinciding with midday and evening hours. The night-time maxima pattern is consistent throughout the week, with the higher concentrations being recorded on Friday and Saturday evenings (Fig. 6b), in line with an expected food service sector activity increase as part of Nicosia inhabitants' leisure in the weekend. The mass spectrum of HOA-2, even though left unconstrained, is highly correlated to COA found in other studies (Fig. S10b) in both Mediterranean and continental European urban environments. Additionally, the non-negligible signal at m/z=60 points to the widely spread habit of meat charbroiling (Kaltsonoudis et al., 2017).

Influence of power plant emissions: A closer look at the diurnal variability of the HOA-2 factor shows a certain persistence of this factor throughout the day, even when cooking activities are more or less absent (Fig. 6a). Such pattern could imply the influence of other combustion sources, not necessarily of local origin. The influence of other combustion sources would also help to explain why HOA-2 average concentrations are roughly 3 times higher than OA related to traffic (HOA-1), as it is very unlikely that cooking activities can contribute solely to the observed HOA-2 concentrations. A possible contributing source could be related to the energy production sector on the island, which relies exclusively on heavy fuel oil. In a recent study, Vrekoussis et al. (2022), utilizing satellite observations, have identified that power plants located to the North (Teknecik powerplant, PP4, 362MW), North-East (Kalecik powerplant, PP5, 153MW) and South-East (Dhekelia power station, PP3,

460MW) of Nicosia at 22 km, 60 km and 38 km, respectively, are significantly contributing to columnar $NO_2$ concentrations
over the island. The importance of these emission hotspots, along with their location on the island, during both the cold and
warm periods is illustrated in Fig. S13 and shows, particularly for the Northern power plants (PP4, PP5), emissions as high as
the traffic-related $NO_2$ over Nicosia. Interestingly, in a source apportioning study on VOCs performed at the Cyprus
Atmospheric Observatory – Agia Marina Xyliatou (CAO-AMX), a rural remote site 32 km southwest of Nicosia, Debevec et
al. (2017) have resolved a factor related to industrial activity/power generation, exhibiting a connection with winds arriving
from the wider eastern sector.
In order to assess the possible influence of Cypriot power plant emissions, the coupling of wind velocity, and wind direction
with the HOA-2 time-series was performed through NWR analysis (Fig S14b). This analysis highlights the association of
stagnant conditions (low wind speed / low dispersion) with high HOA-2 concentrations (i.e., night-time peaks), pointing to a
more local origin for this OA source. On the other hand, different features appear when wind velocities are higher, showing
emissions originating from the NW and the E-NE sectors; i.e. downwind of power plants PP4 and PP5, although long-range
transport influence cannot be ruled out. This is illustrated by the NWR of sulfate (Fig. S14f), which shows a dominant E sector
likely to originate from regional emissions. Given the positioning of the sampling site, close to the edge of Nicosia's urban
fabric, with the Athalassa park lying to the east, such an observation can suggest the transport of plumes from the operating
powerplants, namely PP5 and PP3 to the city. Interestingly, a similar yet even clearer image stands for $SO_2$ concentrations –
only half of which are considered to be of urban origin (Vrekoussis et al., 2022) – measured at a suburban background site
(NicRes) and a traffic site (NicTra) in the city (Fig. S14g-h), with elevated $SO_2$ concentrations being related to eastern winds
of higher velocity, further corroborating that power generation related polluted plumes, traveling through the Mesaoria plain
arriving to Nicosia can contribute to the HOA-2 factor.
Other combustion sources: Interestingly, chloride shows a good correlation with HOA-2 ($r^2$=0.61; N=2945; see Fig. S11f, Fig.
5b). Chloride detected by the ACSM is in the form of $NH_4Cl$ (a secondary highly-volatile species). The source of this chloride
is still widely debated and may originate from industrial activity or municipal (plastic-containing) waste burning (Gunthe et
al., 2021). Another possible explanation of the good agreement between HOA-2 and chloride would be the use of Cl-rich coal
as a means for outdoor cooking in Nicosia could therefore reflect the influence of cooking activities that comprises a fraction
of the HOA-2 factor.
**Less-Oxidized Oxygenated OA (LO-OOA):** With elevated contribution of m/z 44, the mass spectrum of this factor is
consistent with a secondary OOA source. A higher m/z 43 and a lower m/z 44 (Fig. 5a) compared to MO-OOA implies a less
oxygenated (less-processed) component (Mohr et al., 2012). Finally, the time series of this factor is quite similar to $NO_3^-$, with
an overall good correlation value ($R^2 = 0.67$, N=2943; Fig. S11h), highlighting its semi-volatile character. This is further
corroborated by the very good correlation of LO-OOA with chloride ($R^2 = 0.73$, N=2943; Fig S11i), another semi-volatile
compound measured by the Q-ACSM. The diurnal variation of LO-OOA displays 1.5 times higher concentrations during the
night compared to daytime (maximum of $1.84 \pm 0.31$ µg m$^{-3}$ at 22:00 LT; Fig. 6a); a pattern that is much more pronounced
than the variability observed for MO-OOA. This feature highlights that the presence of LO-OOA, is not exclusively controlled
by photochemical processes. Instead, changes in thermodynamic equilibrium (due to lower T and increased RH), favouring
the condensation of gas-phase semi-volatile material on the one hand, and intense night-time chemistry (gas phase or
heterogenous) on the other hand, are among the processes that may account for the rapid night-time formation of LO-OOA.
Atmospheric processing of biomass burning OA during periods of low photochemical activity (such as in winter or at night),
also known as "dark" aging, has been reported recently (Kodros et al., 2020; Jorga et al., 2021) and could have contributed to
the observed night-time formation of LO-OOA. Notably, the weekly cycle of LO-OOA, and its night-time maxima, appears
to have the same pattern and intensity as those observed for BBOA (e.g., low peaks on Tuesday/Thursday, maximum on
Sunday) (Fig. 6b). On the other hand, the factor is correlated with both BBOA ($R^2$=0.81; Fig. S11k) and $BC_{wb}$ ($R^2$=0.66; Fig.
S11j). This observation could indicate a biomass-burning contribution to LO-OOA through fast oxidation of primary
emissions, supported by several studies showing biomass burning linked to OOA sources at night (Stavroulas et al., 2019;
Kodros et al., 2020; Chen et al., 2021).
**More-Oxidized Oxygenated OA (MO-OOA):** The MO-OOA factor typically accounts for secondary organic aerosol formed
in the atmosphere from gas-to-particle conversion processes of VOCs and their products, as well as atmospheric ageing of
primary OA (Petit et al., 2015; Stavroulas et al., 2019). Numerous VOC sources can contribute to OOA but lose their mass
spectrum fingerprint owing to extended oxidation due to photochemical aging, which leads to enhanced signal at the m/z 44
fragment ($CO_2^+$), a dominant tracer for OOA (Ng et al., 2011). The predominance of m/z 44 and the near absence of m/z 43 in
the mass spectrum of the resolved MO-OOA factor (Fig. 5a) points to highly oxidized/aged secondary OA (i.e., originating
from long-range transport). This is further supported by the relatively good agreement ($R^2$=0.55; N=2943; Fig. S11l) between
concentrations of MO-OOA and sulfate (Fig. 5b), a species of regional origin (Sciare et al., 2003). Nevertheless, the diurnal
variability of MO-OOA does not closely follow sulfate showing a small increase of 20-30% every evening (Fig. 6a,b), which
furthermore cannot be explained by atmospheric dynamics (c.f. the negligible PBLH diurnal variability for the cold period
shown in Fig. S8). Alternatively, this would suggest that a fraction of MO-OOA is produced locally through night-time
oxidation mechanisms as previously observed for LO-OOA. Similar nighttime increases of high oxygenated OA factors,
related to local sources, have been reported in both northern European urban sites (Zhang et al., 2019; Lin et al., 2022) as well
as in the Eastern Mediterranean urban environment (Athens, Greece), where a link to oxidized primary residential wood
burning emissions as a potential driver of the low volatility OOA factor diurnal variability, was also suggested (Stavroulas et
al., 2019).

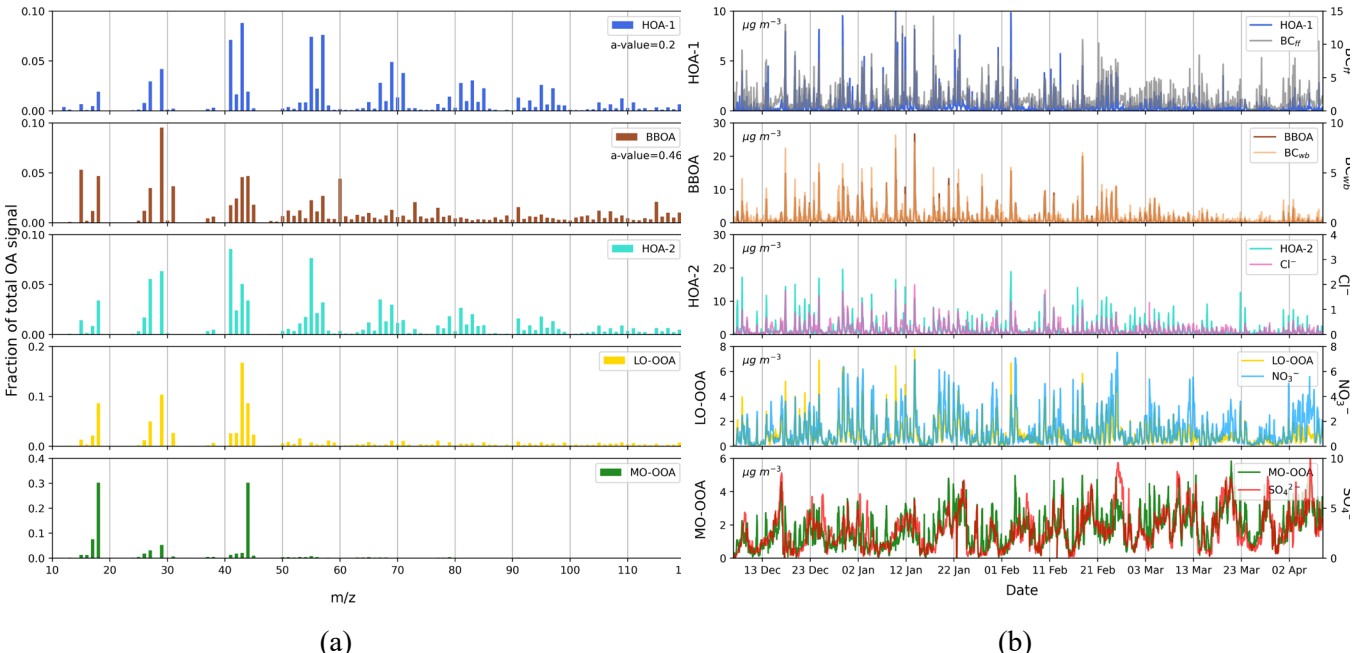

(a)  (b)

**Figure 5: Mass spectra of the PMF (a) and time series of the five OA factors resolved along with corresponding tracer compounds**
**(b) for the cold period.**

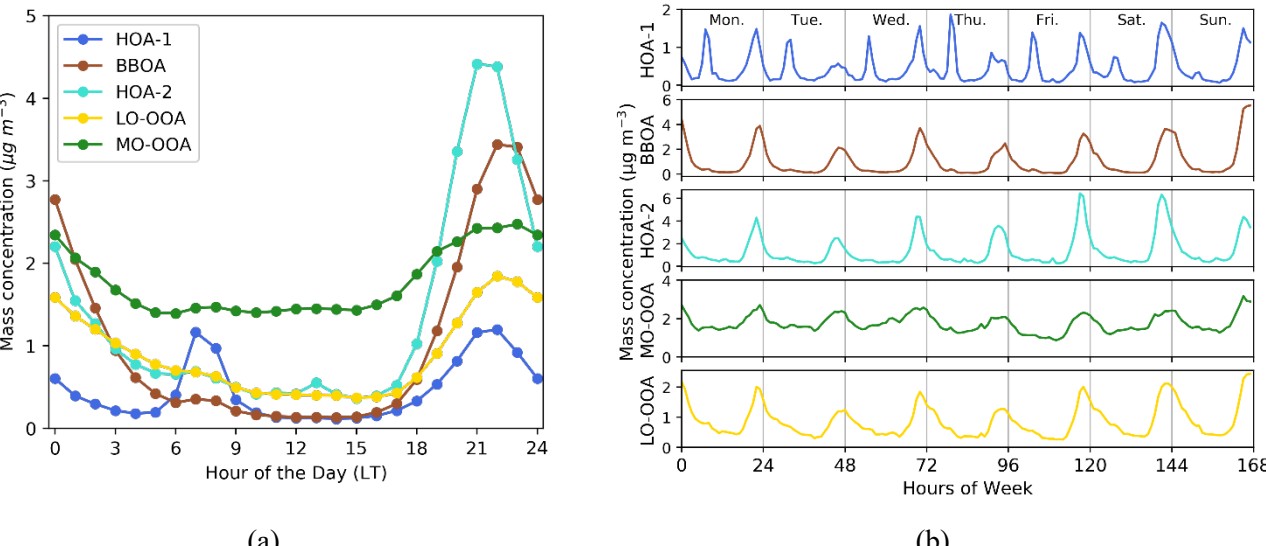

|(a)|(b)|

**Figure 6: Diurnal variability (left) and weekly cycles (right) of the five OA factors averaged over the cold period..**
**3.5.2. OA source apportionment during the warm period**
For the warm period, the optimal PMF solution was obtained using a 4-factor solution (HOA-1, HOA-2, MO-OOA, LO-OOA).
As expected, the BBOA factor could not be resolved, as previously highlighted by the low concentrations at m/z 60 reported
during this period (Fig.2). Again, the identification of OA sources related to the 4 OA factors was performed following the
typical combination of information from i) OA mass spectra (Fig. 7a), ii) the correlation of each factor with external source-
specific tracers (Fig. 7b and Fig. S15), iii) their diurnal variability (Fig. 8a), and iv) their daily (weekdays vs. weekend) pattern
(also Fig. 8b). The mass spectra profiles for the 4-factor PMF solution during the warm period (Fig. 7a) were quite similar to
the ones from the cold period (Fig. 5a).
**HOA-1:** For the warm period, an a-value of 0.2 was selected for constraining the HOA-1 factor, again using the Ng et al.
(2011b) HOA profile as a reference. The resolved factor profile is nearly identical to the one obtained for the cold season ($R^2$
= 0.99, Fig. S10a). It is also very well correlated to traffic-related HOA factor profiles found in other Mediterranean
(Kostenidou et al., 2015; Gilardoni et al., 2016; Florou et al., 2017; Stavroulas et al., 2019) and European cities (Lanz et al.,
2010; Crippa et al., 2014) as depicted in detail in Fig S10a. The HOA-1 time series follows the same pattern as the
corresponding traffic-related HOA-1 factor reported for the cold period, showing a good correlation with $BC_{ff}$, ($R^2$=0.62,
N=1259; Fig. S15a). Its diurnal variability exhibits a bimodal pattern, with a typical sharp maximum in the morning (07:00
LT) and a smaller peak during the evening (Fig. 8a). On a weekly basis, this diurnal variability tends to be less pronounced on
Saturdays and nearly absent on Sundays (Fig. 8b), reflecting reduced commuting during the weekend.
**HOA-2**: The HOA-2 factor still shows elevated concentrations during the warm period, close to 3 times higher than HOA-1
(Table S3). The profile remains quite unchanged between the cold and warm periods ($R^2$= 0.92; Fig. S10b), pointing to similar
sources. No correlation was observed with chloride, which may be expected due to unfavourable thermodynamic conditions
hindering $NH_4Cl$ formation as well as the lack of significant chloride sources during this period. A night-time maximum of
HOA-2 is still observed when investigating the factor's diurnal variability (Fig. 8a). Furthermore, a somewhat broader,
compared to the cold period, maximum in the middle of the day (Fig. 8a) can also be observed. When going through the weekly
variability, this midday maximum is particularly well defined on Sundays (Fig. 8b), while the evening peaks of Sundays and
Mondays are the lowest. The above observations remain consistent with the cold period assessment, that HOA-2 is on the one
hand linked to cooking activities. For household activities are expected at noon and evenings, while for restaurants, activity
peaks on Sunday noon and is lower on Sunday evening and Monday, reflecting the fact that such businesses remain closed on
the first day of the week (Fig. 8b). On the other hand, the overall offset of HOA-2 observed against the HOA-1 diurnal profile
persists, suggesting somewhat permanent background HOA-2 concentrations that cannot be explained by cooking activities
alone. A contribution to this source by continuous emissions from power plants (see space-based (SP5-TROPOMI) vertical
columns of $NO_2$ during the warm period in Fig.S13d) should be sought. In addition, the HOA-2 NWR plot for the warm period
reveals an even more significant enhancement of concentrations when moderate winds blow from the E-SE (Fig. S16b), a
trend also observed for $SO_2$ during the same period (Fig S16e,f).
The above observations remain consistent with our assessment for the cold period: the HOA-2 factor consists of a mixed OA
source that contains cooking activities (inc. coal combustion) and emissions from the powerplants located on the eastern part
of the island. Indeed, the HOA-2 midday maximum can be linked to an increase in electricity demand at that time of day during
the warm period due to an increase in air conditioning usage (Cyprus' NECP 2021-2030, 2019).
**LO-OOA:** The LO-OOA factor profile exhibits some differences from the one resolved for the cold period ($R^2 = 0.66$), as
illustrated in the correlation matrix of comparison to selected factor profiles found in the literature (Fig. S10d) while being
very similar to those obtained in Athens/Piraeus during summer (Bougiatioti et al., 2014; Stavroulas et al., 2021). The LO-
OOA time-series shows a low agreement with $NO_3^-$ ($R^2 = 0.31$; N=1259; Fig. S15c) poorer than the observed correlation
during the cold period (Fig. S11h). The diurnal pattern of the factor (Fig. 8a) shows maximum concentrations persisting
throughout the night and early morning, while a secondary maximum during the midday can be observed. But overall, the
diurnal pattern of LO-OOA is rather flat compared to the cold period, suggesting that local production may not be so important
at that time compared to a less variable regional background. Interestingly a midday hump similar to the one observed for
HOA-2 is present, suggesting a common origin.
**MO-OOA:** The factor profile of MO-OOA resolved during the warm period is strikingly identical to the profile found in the
cold period (their $R^2$ is almost 1; Fig. S10e), while being excellently correlated to other highly oxygenated OA factors resolved
in both the urban and regional background in the Eastern Mediterranean (Bougiatioti et al., 2014; Stavroulas et al., 2019, 2021)
as well as in continental Europe (Crippa et al., 2014). The winter night-time peaks are not observed anymore (Fig. 8a), with
the factor's diurnal pattern exhibiting much less variability, highlighting its dominant regional character. The time series of
MO-OOA correlates well to $SO_4^{2-}$ ($R^2$=0.53; N=1259; Fig.S15b), confirming this regional and highly processed origin. The
concentration levels of MO-OOA during the warm period are lower than in the cold (Table S3). However, its relative
contribution to total OA during the warm period remains similar (45 %).

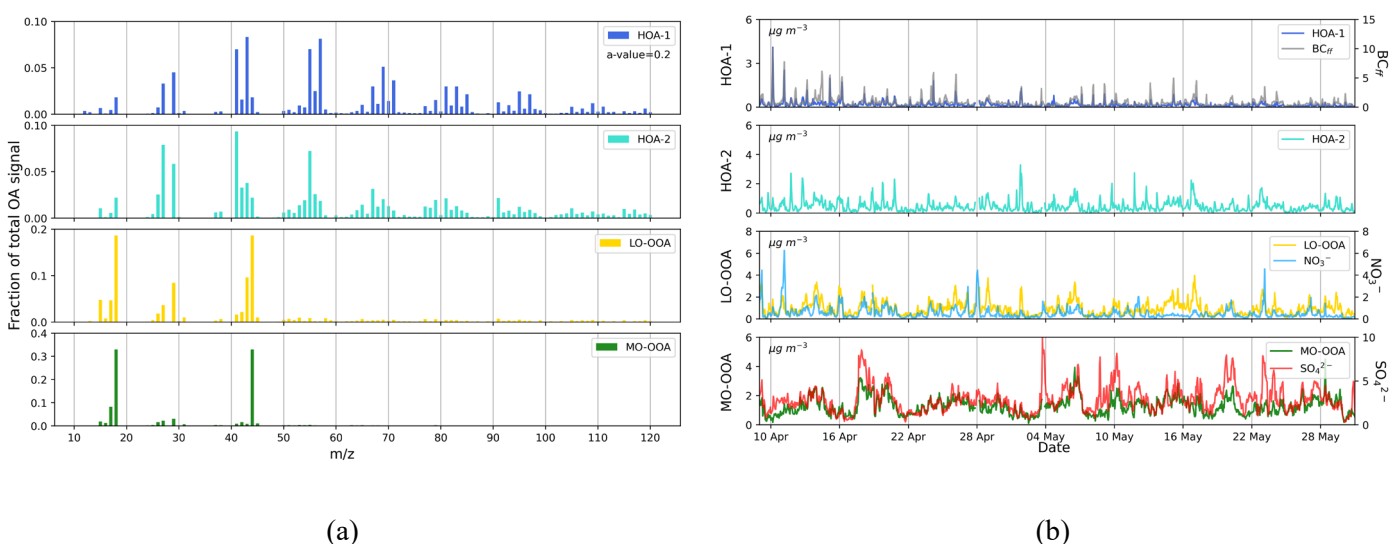

(a)                                                                            (b)

**Figure 7: Mass spectra of the PMF (a) and the time series of the four OA factors resolved along with corresponding tracer compounds**
**(b) for the warm period.**

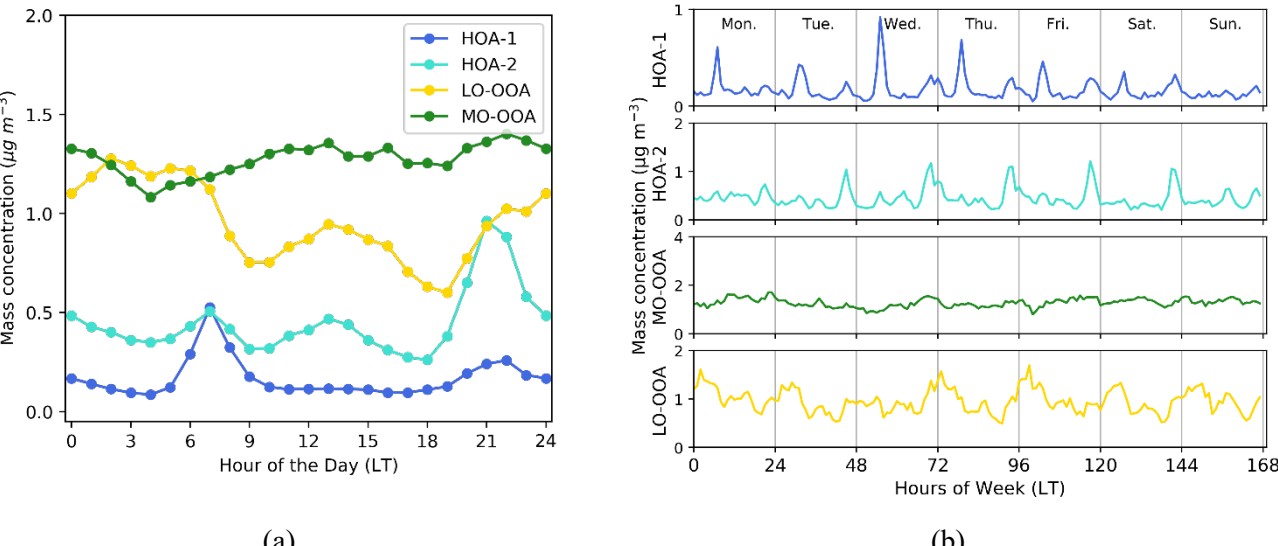

(a)

(b)

**Figure 8:** Diurnal variability (a) and weekly cycles (b) of the four OA factors resolved during the warm period.

**3.6. Spatial and seasonal variability of OA sources**

**3.6.1. Seasonal variability of OA sources**

**Primary OA:** The mass concentration of the three primary OA factors (HOA-1, HOA-2, BBOA) represents as much as 40 % of total organic aerosols during the cold period (Fig. 9), with POA contribution significantly decreasing in the warm period (22% to total OA) due to the absence of the significant residential wood burning source which during the cold period accounted for 12% of total OA. The important contribution of primary sources in Nicosia has also been highlighted earlier by the rather low OA/OC ratio of 1.42 (Section 3.1). In a recent publication covering several European sites, Chen et al. (2022) reported that in urban sites, solid fuel combustion-related OA components were 21.4 % of total OA during winter months, higher than what is found for BBOA in Nicosia, owing to the rather milder winters in the city.

The traffic-related primary factor in Nicosia (HOA-1) was found to be rather stable in terms of contribution to total OA across this study's two seasons, averaging 7% and 6%, respectively, for the cold and warm periods, being lower than the figure reported in other European Urban sites (12.7%, Chen et al., 2022). On the other hand, the HOA-2 factor represents ca 2/3 of the total HOA in Nicosia with little variation from winter (72 %) to summer (66 %) to total HOA (Fig. 9). Comparing it with COA in urban locations resolved by Chen et al. (2022), during both winter (14.4% compared to 21% in the cold season in Nicosia) and spring (15% versus 16% in Nicosia during the warm season), the higher values reported in Nicosia further support the assumption that the HOA-2 represents a mixed combustion source.

**Secondary OA**: A higher degree of oxidation is observed for the LO-OOA factor during the warm period, given the much higher signal contribution at m/z 44 than the respective cold period factor. This discrepancy, reported in several studies (Huang et al., 2019; Duan et al., 2020), is explained by higher photochemistry during the warm period, which promotes the oxidation of OA, resulting in a LO-OOA profile with a higher m/z 44 fraction. This result is also consistent with a less-oxidized LO-OOA formed during the cold period from night-time chemistry. The range of LO-OOA concentration levels is different between cold and warm periods (0.05-7.74 µg m$^{-3}$ and 0.05-4.00 µg m$^{-3}$, respectively), while the mean concentrations for both periods are similar (0.86 and 0.95 µg m$^{-3}$ for cold and warm periods respectively). The contribution of LO-OOA relative to total OA is double during the warm period compared to the cold, reflecting both the absence of the biomass burning source as well as the prevailing conditions favoring atmospheric processing of primary OA and SOA precursors. During the cold period, LO-OOA intense peaks suggest an influence from local emissions, while during the warm period, the less-variable LO-OOA diurnal variability highlights the influence of more intense photochemical processing at medium-to-large geographical scale. MO-OOA is found to be the major contributor to total OA for both the cold (44%) and warm (45%) periods, higher in both

cases than the average MO-OOA contributions reported for other European urban sites (Chen et al., 2022) underlining the
importance of highly processed secondary OA over Nicosia (Fig. 9).

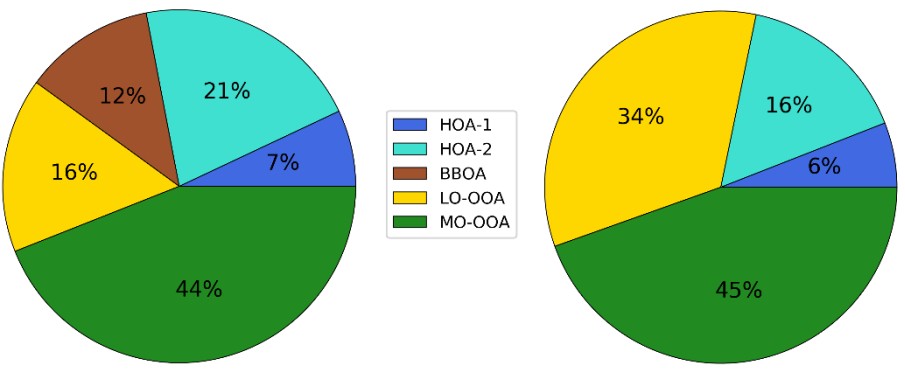


**Figure 9: Relative contribution of PMF resolved OA sources to total OA for the cold period (left) and the warm period (right), respectively.**

### 3.6.2. Geographic origin of OA sources

The geographic origin of OA sources (local vs regional) is further assessed here using both Non-parametric Wind Regression
(NWR) analyses as well as the regional scale coupling concentrations to air mass back trajectories through PSCF.
**Cold period:** During this period, primary OA factors, especially HOA-1 and BBOA, have an expected strong local component
that is characterized by high concentrations at low wind speeds (hourly average 1.4 m s$^{-1}$) when winds are originating from the
W-SW sector (Fig. S14a,c), pointing to the busy highway connecting Nicosia to the other major cities in the island while
integrating the highly populated residential areas of Strovolos and Lakatamia municipalities. (Fig. 1c). As discussed earlier,
the HOA-2 factor, apart from its local influence (also in the W-SW sector), exhibits significant concentrations related to higher
wind speeds from the NW and the E-NE sectors that could originate from power plants but also possibly from long-range
transport. Interestingly, a small local contribution from the city, still within the W-SW sector, can also be observed for both
LO-OOA and MO-OOA, consistent with the peaks observed that could originate from local night-time chemistry. Still, high
concentrations of MO-OOA (and, to a lesser extent LO-OOA) are observed with high wind speeds and Eastern directions (Fig.
S14e,d). Although the contribution of the power plant PP5 located in the East sector (Fig. S13c) cannot be excluded, PSCF
analysis points out that the hotspots of MO-OOA can be traced in neighbouring countries (eg. Syria, Lebanon and South
Turkey) in the middle East (Fig 10a). These areas also represent hotspots of SO$_4^{2-}$ according to PSCF analysis (Fig. S17a).
**Warm period:** Given the generally higher wind speeds recorded, in comparison to the cold season (average of 1.93 m s$^{-1}$ vs.
1.36 m s$^{-1}$ in the cold period), all OA factors show elevated concentrations coupled with higher wind speeds. The most striking
result is the major influence of the E-SE sector for all OA sources. However, this sector is upwind of Nicosia and, therefore,
poorly influenced by local city emissions. As noted previously, for the cold period, long-range transported OA from the Middle
East is expected to be the main driver to explain the influence of the E-SE sector, at least for LO-OOA and MO-OOA (Fig.
S16c,d). This is again confirmed by the PSCF results reported in Fig. 10b for the warm period. The HOA-1 factor still shows
maxima for low wind speeds (<5 km h$^{-1}$) characteristic of local emissions and the SW-S direction but also exhibits significant
contribution related to the E-SE sector. Although the influence of the power plant PP5 on HOA-2 is expected, the contribution
of this source can not be excluded for HOA-1 as well. On the other hand, quantification of the Middle Eastern contribution to
the HOA-2 factor remains to be assessed since the current dataset cannot provide sufficient information on separating the
contribution of power plants on the island versus more regional Middle East emissions (Fig. S165b). Although this hypothesis
needs further investigation, the presence of HOA-2 in the Middle East would be consistent with recent findings highlighting
the importance of OC emissions from diesel generators used in Lebanon as a means of complementary power generation (Fadel
et al., 2022).

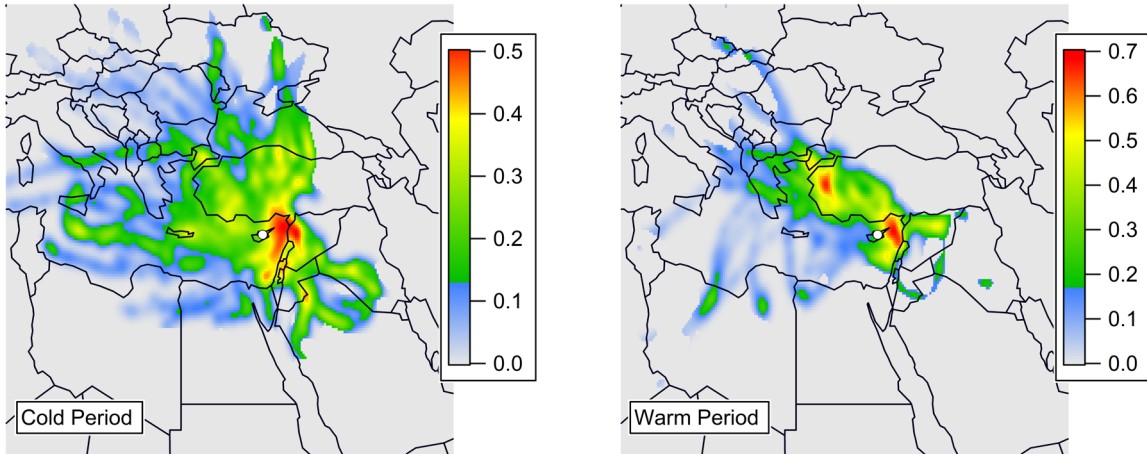


**Figure 10: PSCF plots for MO-OOA during the cold and warm periods. The color scale represents the probability of air parcels arriving at the receptor site (white dot) for measured concentrations higher than the 75th percentile.**

In conclusion, based on the relative contribution of OA factors (Fig. 9) and the NWR analysis (Fig. S14, S16), it can be
reasonably assumed that a significant amount of measured OA in Nicosia originates from long-range transport with the Middle
East being the major source region, during both cold and warm periods. This is the first time that such a high contribution of
OA from the Middle East is highlighted over Cyprus. Assuming that biomass combustion and biogenic emissions of OA in
the desert regions of the Middle East are relatively limited, these results suggest that most of the primary and secondary OA
originating from the Middle East could be of fossil fuel origin, which is consistent with the previously reported extensive use
of oil in this region.
**3.7. Spatial and seasonal variability of BC sources**
The above conclusion on the influence of primary and secondary OA sources from the Middle East region, and its strong fossil
fuel origin, motivates a careful examination of the geographic origin and sources of BC concentrations recorded in Nicosia.
Baseline (i.e., lowest) $BC_{ff}$ concentrations are typically observed in the middle of the night and in the middle of the day when
local emissions are at their minimum (See Fig. 4). As such, these background concentrations can be considered as a first
qualitative indicator of background $BC_{ff}$ concentrations of regional origin. Interestingly, these baseline $BC_{ff}$ concentrations
appear to be in phase with those of sulfate (Fig. 11), as well as the MO-OOA factor derived from the OA PMF analysis. This
observation points to the possible use of MO-OOA as a tracer for regional $BC_{ff}$. Hence, it brings further evidence of the
importance of regional emissions on carbonaceous aerosol concentrations in Nicosia.

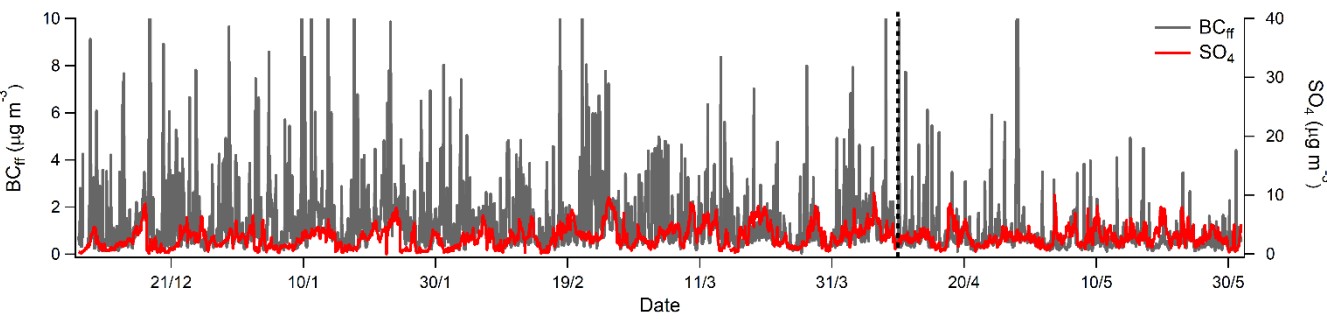

**Figure 11: Temporal variability of $BC_{ff}$ and $SO_4^{2-}$ concentrations during the entire measuring periods.**

The assumption that transported regional pollution can affect $BC_{ff}$ concentrations in Nicosia can be further supported by
investigating the $BC_{ff}$ NWR polar plots for both the cold and warm seasons (Fig. S18a,b). Elevated concentrations related to
local emissions were observed for calm conditions with low wind speeds (<5 km h$^{-1}$) in the SW sector, as previously observed
for HOA-1. Interestingly, $BC_{ff}$ NWR plots show a distinct contribution at higher wind speeds (~15 km h$^{-1}$) and the NE-SE
(Middle East) sector, during both the cold and warm periods, with estimated concentrations of roughly 1.5 μg m$^{-3}$, further
support the major role of the Middle East in the observed BC concentration levels in Nicosia (Fig S18 a,b).
BC source apportionment: In order to better assess the relative contributions of the multiple primary OA sources (HOA-1,
HOA-2) and to quantify the contribution of long-range transport from the Middle East to $BC_{ff}$, a multilinear regression (MLR)
model was tentatively performed using the principle of co-emission of $BC_{ff}$ and organic species by the different sources
(Chirico et al., 2010; Laborde et al., 2013). This approach, used recently by Poulain et al. (2021), assumes that at any given
time (t), $BC_{ff}$ mass concentration is the sum of BC from traffic (traced by HOA-1), from a mixed combustion source (traced
by HOA-2), and from long-range transport (traced by MO-OOA), as follows:
$$[BC]_{ff} = [BC]_{traffic} + [BC]_{mix\ combustion} + [BC]_{regional}\ (\ 2)$$

With:
$$[BC]_{traffic} = a \times [HOA-1]\ (3)$$

$$[BC]_{mix\ combustion} = b \times [HOA-2]\ (4)$$

$$[BC]_{regional} = c \times [MO-OOA]\ (5)$$

Where a, b and c are coefficients derived from the multi-linear regression model.
The above approach assumes that primary HOA-1 and HOA-2 can trace $BC_{traffic}$ and $BC_{mix\ combustion}$, respectively. This is
expected for traffic with a typical HOA-1/$BC_{traffic}$ ratio with little variations. For HOA-2, this assumption is valid for the
fraction assumed to originate from power plant emissions and for some of the cooking activities (e.g., when using charcoal
combustion) but not necessarily all. As such, the uncertainties of this approach are expected to be higher for HOA-2 compared
to HOA-1. The use of MO-OOA to trace the regional source of BC would probably lead to even higher uncertainties because
MO-OOA is also sensitive to atmospheric photochemical processes and does integrate multiple sources. Nevertheless, this
latter assumption is believed to be acceptable given the good agreement reported above between baseline concentrations of
$BC_{ff}$ and MO-OOA (Fig. S19) and the above conclusions that carbonaceous aerosols originating from the Middle East are
expected to be dominated by fossil fuel combustion. Note that MO-OOA was preferred here to LO-OOA to trace regional
emissions due to the latter's somewhat more local character.
Combing equations 2-5 provides the multilinear regression model with the free regression parameters a, b, c, which are fitted
to the time-resolved $BC_{ff}$ mass concentration measured by the Aethalometer and PMF results for the ACSM data:
$$[BC]_{ff} = a \times [HOA-1] + b \times [HOA-2] + c \times [MO-OOA]\ (6)$$

Previous studies have shown that MLR models have enhanced explanatory power when primary emissions dominate (Laborde
et al., 2013). To reduce this potential bias, the MLR model was applied distinctly for the two seasons separately.
During the cold period, a very good correlation between measured and modelled $BC_{ff}$ was obtained (r² = 0.70; N = 2942), with
the modelled $BC_{ff}$ explaining 84 % of the measured one (Fig. S20a). The regression coefficients $a$ (HOA-1), b (HOA-2) and
c (MO-OOA) were found to be 1.11 ± 0.03, 0.15 ± 0.01 and 0.41 ± 0.01, respectively. Regarding the warm period, it was not
possible to obtain a positive value for b (HOA-2). A correlation between long-range transported HOA-2 and MO-OOA is,
among other, a reason that can be proposed to explain why it has not been possible to extract a $BC_{mix\ source}$ factor here. Therefore,
$BC_{ff}$ was only apportioned using HOA-1 and MO-OOA. A good correlation between measured and modelled $BC_{ff}$ was
obtained (r²=0.62; N=1251), with the modelled $BC_{ff}$, explaining 83% of observations (Fig S20b). The regression coefficients
$a$ (HOA-1) and c (MO-OOA) were found to be 3.05 ± 0.07 and 0.19 ± 0.01, respectively.
The combination of the Aethalometer model (apportioning $BC_{ff}$ and $BC_{wb}$) and the MLR model (apportioning $BC_{traffic}$, $BC_{mix}$
$_{source}$, and $BC_{regional}$) was performed to obtain an integrated picture of BC sources in Nicosia for both periods (see Fig. 12).

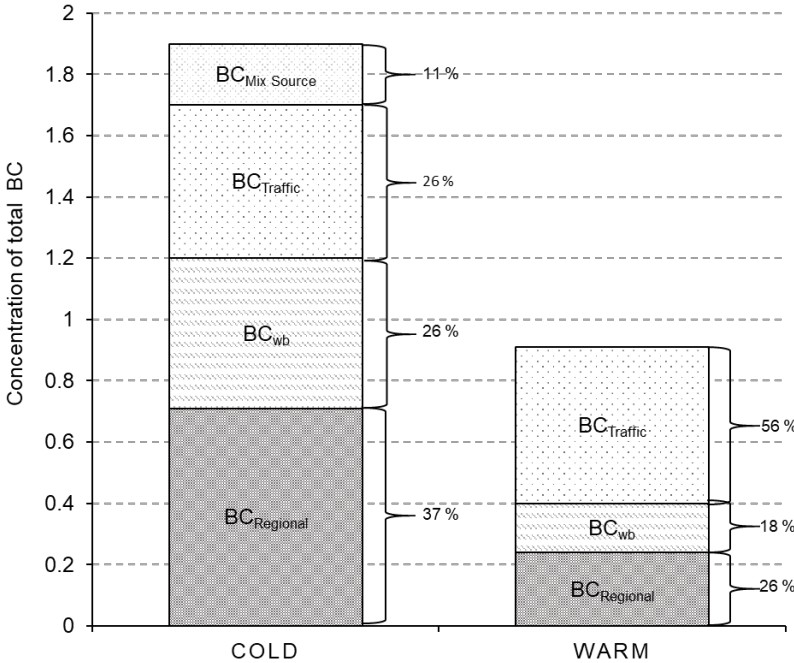

**Figure 12:** BC sources during the cold and the warm period in Nicosia

Spatial and seasonal variability of BC sources: During the cold period, BC was found to originate from four different sources denoting the complexity of combustion sources of different origins in Nicosia. $BC_{regional}$ is the dominant source of BC (37%), while traffic, wood burning, and mix source are estimated to contribute to 26 %, 26% and 11% of BC, respectively. From the perspective of $BC_{ff}$ sources, long-range transport, traced by MO-OOA, remains the largest source of $BC_{ff}$ during the cold period, contributing 63 %, while $BC_{ff}$ from local emissions constrained with HOA-1 and HOA-2 represents 24% and 13%, respectively (Fig S21). In other words, more than half of $BC_{ff}$ in Nicosia was found to be regional and probably originated from the Middle East during the cold period. This high contribution of regional $BC_{ff}$ is quite unexpected for a medium-sized European city like Nicosia, where local traffic is likely to be the main contributor to $BC_{ff}$.

Nevertheless, extra caution should be taken here. The obtained contribution of 63% for $BC_{ff}$ regional should be seen as an upper limit since a fraction of MO-OOA was shown to be of local origin during the cold period. During the warm period, the picture remains similar, with traffic and wood burning representing two-thirds of BC (56 % & 18 %). Here, BC regional contributed 26 % to total BC. From the perspective of $BC_{ff}$ sources during the warm period, the long-range transport contributed 41 %, while $BC_{ff}$ from local emissions constrained with HOA-1 represents 59 % (Fig S21). Although the two models (Aethalometer and MLR) are associated with non-negligeable uncertainties, the BC source apportionment obtained shows that local emissions cannot be considered only for BC, with demonstrated significant contribution of Middle East fossil fuel emissions.

## 4. Conclusions

Near-real-time chemical composition of submicron aerosols and source apportionment of carbonaceous aerosols was performed for the first time in Nicosia, a medium-sized European capital city (circa 250,000 inhabitants) in Cyprus located in the Eastern Mediterranean and surrounded by Middle East countries with fast-growing population and increasing emissions of air pollutants. Continuous observations were performed at an urban background site for approximately 6 months (between 7 December 2018 and 31 May 2019) in order to obtain a large and representative dataset capturing specific features - related to both the cold and warm periods - such as domestic heating and regional transport. Measurements of the major fractions of $PM_1$ were carried out with a Q-ACSM and an Aethalometer complemented by a comprehensive suite of collocated instruments (e.g., filter sampling, SMPS) to assess the quality of the acquired data further.

Unlike many European cities, no clear $PM_1$ pollution episodes of several consecutive days could be observed over Nicosia.
However, very intense peaks (above 40 µg m$^{-3}$, 1h averages) were recorded systematically every evening during the cold
period. Carbonaceous aerosols (BC and OA) were identified as the main components of these peaks and were mostly attributed
to local emissions from heating with little contribution from local meteorology (PBL height did not show significant diurnal
variability during the cold period). Furthermore, a significant portion of $PM_1$ was found to be related to long range transported
aerosol, while the influence of shipping emissions was estimated to be rather low (less than 8%).
Source apportionment of OA has been used to derive a local biomass burning OA ($BBOA_{cy}$) mass spectrum in order to
apportion the contribution of domestic wood burning properly. A total of five OA sources were identified during the cold
period, among which four are typically reported within urban environments (HOA-1, BBOA, LO-OOA, MO-OOA). An
additional one (HOA-2) was assigned as a mixture of several combustion sources, such as cooking as well as a significant
contribution from power plants located in the Northern part of the island. These power plants in addition, represent major
island-based hotspots of $NO_x$, as evidenced by satellite observations. Interestingly, a similar HOA-2 source was identified at
our regional background site (40 km distance from Nicosia; Chen et al., 2022), pointing to a possible influence from these
power plants to an extended part of the island. The impact of this specific source brings the OA contribution of primary sources
up to 40 % over Nicosia during the cold period. Few additional features were noticed for the other OA sources with 1) a typical
traffic-related (HOA-1) source observed during both seasons, 2) a biomass burning source (BBOA) related to domestic heating
enhanced at night during the cold season and accounting for 12 % of the total OA, 3) a less oxidized secondary (LO-OOA)
source of a semi-volatile character, influenced by local night-time chemistry, that was more oxidized (i.e., of a less local
character) during the warm period, and 4) a secondary (MO-OOA) source mostly of regional origin but also influenced by
night-time chemistry during the cold period.
The geographic origin of each OA source was assessed for both seasons. Except for MO-OOA, which systematically shows a
strong regional component, HOA-1, HOA-2, and LO-OOA exhibit a clear local origin during both seasons, and a more
pronounced influence from the Eastern wind sector during the warm period. The prevalence of this sector is systematically
observed for MO-OOA highlighting the major role of Middle East emissions in contributing to almost half of OA
concentrations in Nicosia during both cold and warm seasons.
To further elucidate the influence of this complex mixture of OA sources on BC levels, source apportionment of BC was
performed by combining i) the aethalometer model to separate BC into its fossil fuel ($BC_{ff}$) and wood burning components
($BC_{wb}$), and ii) a multi-linear regression model to apportion the contribution to $BC_{ff}$ from traffic (constrained by HOA-1), mix
combustion sources from cooking and power plants (constrained by HOA-2), and long-range transport from the Middle East
(constrained by MO-OOA). Although several assumptions and uncertainties are associated with this approach, it has shown to
provide an interesting tool for reconstructing the BC concentrations derived experimentally. Such BC apportionment
performed for both cold and warm seasons solidified the conclusions reached through the OA source apportionment, with
almost half of $BC_{ff}$ being of regional origin, with the Middle East playing an important role. This result is quite unexpected
given that local traffic emissions are usually considered the dominant contributor to $BC_{ff}$ in urban background environments.
These conclusions have numerous implications related to PM regulation and the efficiency of local abatement strategies (in
particular regarding traffic emissions), health (combustion aerosols being considered as particularly adverse for human health),
and climate (major influence of light-absorbing aerosols from the Middle East fossil fuel emissions).
More accurate OA and BC source apportionment i) with more co-located high-resolution measurements of specific trace metal
and organic tracers, ii) better resolved OA mass spectra (e.g., from HR-ToF-AMS), iii) the use of various source-specific mass
spectra fingerprints (e.g., from cooking or power plants), and iv) multi-site measurements (incl. both urban and regional
background) will enable a more accurate estimation of local vs. regional fossil fuel emissions in Cyprus while better
constraining the current regional efforts on air quality modelling and forecasting.

Data availability: All data used in this study can be accessed here: https://doi.org/10.5281/zenodo.7802065. More details on
the analyses are available upon request to the contact author Aliki Christodoulou (a.christodoulou@cyi.ac.cy).
Author contributions. AC, IS, MP, PG, KO, EB, JK, RSE, MI, and MR contributed to the acquisition of measurements. AC,
IS contributed to the processing of the data. AC, IS, and JS wrote the manuscript. MV, NM, MD, SS, and MP contributed to
the scientific review and improvement of the manuscript. All authors have read and agreed to the published version of the
manuscript
Acknowledgements.
This paper contains modified Copernicus Sentinel data processed at IUP Bremen. The authors thank Andreas Richter for
providing the TROPOMI/S5P level 1 and 2 products. This study has been co-funded by the European Union's Horizon 2020
research and innovation programme under grant agreement No 615 856612 (EMME-CARE) and by the Norwegian Financial
Mechanism and the Republic of Cyprus under the ACCEPT project (CY-LOCALDEV-0008) in the framework of the
programming period 2014 – 2021.
Competing interests. The authors declare that they have no conflict of interest. At the time of the research, Matic Ivančič and
Martin Rigler were employed by the manufacturer of the Aethalometer.

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
