# Peer review of "Ambient carbonaceous aerosol levels in Cyprus and the role of pollution transport from the Middle East."

_EGUsphere, 2022_

## Author Comment (AC1)

**We thank the two anonymous reviewers for their efforts and constructive comments that helped improve this manuscript. All comments are thoroughly addressed below. Reviewers' comments are reported in *italics*, our responses are included in blue, and updated manuscript text is included in red. Changes are tracked in the revised manuscript in red.**

Reviewer 1

*This study looks at a unique dataset gathered by advanced aerosol composition measurement techniques from a region not as well studied by online aerosol chemistry and composition. The study adds to the wealth of knowledge of aerosol pollution and long-range transport, especially owing to the unique impacts of sulfate on Cyprus and impact of Power Generation Plants. Several established scientific methods are used to establish pollution sources impacting the site and the authors do a nice job of using multiple techniques to evaluate and come to some conclusions. Generally, a nice paper, although with little novel methods, applied to a new and distinctive dataset. General comments are below with detailed comments following.*

We appreciate the positive comments shared here by the reviewer. Indeed, the presented study does not report on a new methodology. However, we would like to mention that the BC source apportionment approach has been reported in very few papers. The study's novelty emanates from the new and distinctive dataset of carbonaceous aerosols in a region (Middle East)  that, despite being an air pollution hotspot, still has very limited observations available to characterize it.

*Based on the back trajectory cluster analysis alone, it is not convincing that there are enough air masses advecting over the Middle East to draw the conclusions the title suggest. The authors may consider adding in a graph which shows all 72-hr air mass back trajectories during the campaign (the authors use 120 hrs but this is excessive giving the uncertainty associated with HYSPLIT) at 6-12 hr intervals. Perhaps this would give a*

*better idea of the potential influence of the Middle East. Please also include the Log10(n+1) trajectory footprint graphs from Zefir for the PSCF in Figure S17.*

We would like to thank the reviewer for this comment. We believe that, indeed, the section regarding the cluster analysis did not manage to describe the dispersion of air masses adequately. We have elaborated on the clustering analysis findings in the revised main text (Section 3.2). More figures have been added in the revised supplemental material. The analysis denotes that a significant portion of air mass back-trajectories arriving in Nicosia has the origin or passes over the middle East and Turkey.

[Figure]

Figure 1. Calculated 72h back trajectories arriving at the measuring site every 6 hours for both the Cold and Warm periods.

As suggested, plotting all available 72h trajectories (see Fig. 1) – which is also included in the revised supplemental material – and focusing on the last 72 hours, further supports our argument, with a clear portion of back trajectories (25% of the calculated trajectories) are being influenced by the Middle East, especially for the Cold period. Cluster 1 in the cluster analysis presented for the Cold period in the manuscript represents 25% of the calculated trajectories.

In addition to the clustering analysis, the title of this study is further justified by results from the PSCF analysis relating the higher observed concentrations of carbonaceous

aerosols to hotspots in the region and by the multi-linear regression exercise performed on measured BC related to fossil fuel combustion.

*While the authors use NWR and PSCF productively to understand the potential sources of pollutants impacting the Nicosia site, I wonder that there is no mention of shipping emission impact given the high levels of sulfate measured. The sulphate burden over the Mediterranean has been shown to be higher in summer than that over Europe owing to shipping activities (Marmer & Langmann, 2005). While source apportionment techniques have shown a range of effects from ship related sources, from OA fractions of 4.5% in the Mediteranean (Chazeau et al. 2021) to 25% of PM2.5 in Hong Kong (Yau et al. 2013), the ship related burden does not seem to be even considered in the text. The Mediterranean will not become a sulfur emission control area (SECA) (marine fuels S content below 0.1%) until 2025 (COP 2021 under MARPOL (Annex VI), decision 10 June 2022), and has only been under the 0.5% S fuel content since Jan 2020 (IMO-2021, Low Sulfur Regulation), which is after this data 2018-2019 was collected. Therefore, it is strongly suggested that the impact of marine shipping be discussed within the main text as a possible source or reasons given why it is not being considered.*

*Marmer, E., and Langmann, B.: Impact of ship emissions on the Mediterranean summertime pollution and climate: A regional model study, Atmospheric Environment, 39, 4659-4669, https://doi.org/10.1016/j.atmosenv.2005.04.014, 2005.*

*Chazeau, B., Temime-Roussel, B., Gille, G., Mesbah, B., D'Anna, B., Wortham, H., and Marchand, N.: Measurement report: Fourteen months of real-time characterisation of the submicronic aerosol and its atmospheric dynamics at the Marseille–Longchamp supersite, Atmos. Chem. Phys., 21, 7293-7319, 10.5194/acp-21-7293-2021, 2021.*

*Yau, P. S., Lee, S. C., Cheng, Y., Huang, Y., Lai, S. C., and Xu, X. H.: Contribution of ship emissions to the fine particulate in the community near an international port in*

*Hong Kong, Atmospheric Research, 124, 61-72, https://doi.org/10.1016/j.atmosres.2012.12.009, 2013.*

We thank the reviewer for the comments on shipping emissions and their contribution to the sulfate background concentrations in our region. Indeed, shipping emissions are likely to play a role in air pollution in the Mediterranean. However, we would like to emphasize here that the detailed source apportionment of carbonaceous aerosols remains the core of our scientific discussions and findings (as reflected in the title of the paper).

Coming back to the influence of shipping emissions at our sampling site, we have carefully reviewed the references provided by the reviewer (Marmer & Langman, 2005). Although these authors highlight the major contribution of ship emissions on $SO_2$ in the Mediterranean, they also show that $SO_4^{2-}$ concentrations from shipping are mostly close to the main shipping routes (Gibraltar-Suez). We have investigated the contribution of shipping emissions on $SO_4^{-2}$, $SO_2$ and total fine particulate matter ($PM_{2.5}$) near-surface levels in the Eastern Mediterranean region using the WRF-Chem (v3.9.1.1) that simultaneously simulates physical and chemical processes taking place in the atmosphere, thus considering direct and indirect links and feedback between chemical components and atmospheric processes (Grell et al. 2005; Fast et al. 2006). The model has been evaluated in several studies for the Easter Mediterranean (Kushta et al., 2014; Georgiou et al., 2018) and Europe (Berger et al., 2016; Tuccella et. al, 2012).

Following the set-up used in Giannakis et al. (2019) and driven by the EDGAR v.5 anthropogenic emission inventories (Crippa et al., 2019), we performed two annual-long simulations: firstly, including all sectoral emissions in the model (baseline simulation, hereafter referred to as $S_0$) and a second simulation where shipping emissions have been omitted (scenario simulation, $S_1$) to identify the impact of shipping on gaseous and aerosol sulfur-related species concentrations ($SO_2$ and $SO_4^{-2}$) and total $PM_{2.5}$ over the Central and Eastern Mediterranean. The figures below describe the contribution of shipping in absolute terms (Fig. 2a,c,e) and as a percentage (Fig. 2b,d,f) for the $SO_4^{-2}$, $SO_2$, and total $PM_{2.5}$ calculated for each species.

[Figure]

Figure 2: Difference in mean annual modelled surface concentrations of (top row) $SO_4^=$, (middle row) $SO_2$ and (bottom row) PM2.5 in absolute values (left) and percentage (right) between the baseline S0 and no-shipping emissions S1 simulations.

According to these results, the highest impact of shipping on near-ground modelled concentrations of the three species ($SO_4^{-2}$, $SO_2$ and $PM_{2.5}$) was estimated along the central Mediterranean region (yellow grids, west of the Balkans and Greece), as well as a small section south of Greece. The Levantine basin, where Cyprus is located, experiences significantly lower influence under the no-shipping emissions sensitivity test. More specifically, over the East Mediterranean, $SO_4^{-2}$ concentrations are reduced by 0.1-0.3 µg $kg^{-1}$ of dry air over the sea, representing a relative change of about 6-8%, increasing to 8-10% on the eastern borderline covering the Aegean islands and the region south of Crete, and extending to 10-18% over the Central Mediterranean. The contribution of emissions from shipping does not exceed 20% over Eastern Mediterranean except for the area south of Cyprus. It reaches up to 50-60% over the main shipping routes in the Adriatic Sea. Similar is the landscape of contribution to total $PM_{2.5}$ that peaks over the Central Mediterranean at 12-16%. The $PM_{2.5}$ results are comparable to the Viana et al. (2011) review that included several Mediterranean studies with mean annual values varying from 2-14% depending on the applied methodology and location. Specifically for Central Mediterranean locations, Contini et al. (2011) estimated the contribution from shipping emissions to $PM_{2.5}$ to be between 1% and 8%. South of Sicily (isle of Lampedusa), the contribution of ship emissions has been found to account for more than 8% of $PM_{2.5}$ (Becagli et al., 2012). Our modelling results also indicate that shipping affects $SO_2$ concentrations significant to $PM_{2.5,}$ as also stated by Merico et al. (2017).

Regarding the contribution of shipping emissions to carbonaceous aerosols:

The contribution of shipping in the Mediterranean on Black Carbon (BC) concentrations was investigated from model estimates by Marmer et al. (2009) based on three (3) most commonly used ship emissions inventories: 1) EDGAR FT by Olivier et al. (2005), 2) Eyring et al. (2005), and 3) EMEP by Vestreng et al. (2007). Results showed that shipping emissions were contributing to typically 15-25% of BC in the E. Mediterranean, far from the shipping routes (which is the case for Cyprus). A model study of sources of organic aerosols in Europe using CAMx (Jiang et al., 2019) showed that the contribution of "other anthropogenic sources" (gathering shipping, industry, and energy production) on OA

(POA+SOA) was, typically, of the order of 10% during summer and winter in the Eastern Mediterranean region close to Cyprus; Figure 8 at this publication). Based on a simple receptor model, $PM_{2.5}$ source apportionment performed in Nicosia, Achilleos et al. (2016) showed that the contribution from shipping is approximately 8% to $PM_{2.5}$. Most of the transported mass is attributed to $SO_4^{2-}$ with a minor contribution from carbonaceous aerosols. A very similar result was found from a more detailed (Positive Matrix Factorization) $PM_{2.5}$ source apportionment analysis performed in Nicosia in 2018, with heavy oil combustion contributing 7% to $PM_{2.5}$ (Bimenyimana et al., 2023 under review), with the relevant factor containing less than 0.1μg $m^{-3}$ of both OC and EC.

To conclude, while model-based studies estimate the contribution of shipping emissions to be close to 10% for both BC and OA over Cyprus (i.e., at regional background sites), experimental studies suggest that shipping emissions could contribute even lower at the urban (Nicosia) background conditions. Our revised manuscript refers to the above and concludes that "shipping emissions are most probably contributing to the regional background sources of carbonaceous aerosols play a minor role compared to the other sources reported in our studies (i.e., traffic, biomass burning, energy production, etc.).

The points mentioned above were added to the manuscript to address additional points raised by the reviewer (Section 3.3).

 *Title - Title reorder suggestion as LRT is not really impacting the sources but rather a large mixed source itself. 'Carbonaceous aerosol over Cyprus impacted by long-range transported pollution from the Middle East'*

Taking into account the reviewer general comment above, the title has been revised accordingly:

"Ambient carbonaceous aerosol levels in Cyprus and the role of pollution transport from the Middle East."

Specific (minor) comments

*Line 73 – insert of, 'in terms of PM'.*

The following text is now added to the manuscript: "Those studies have highlighted the unique location of Cyprus as a receptor site of major regional pollution hotspots, making the island one of the most polluted EU member states in terms of PM and O₃ concentrations;"

*Line 86 – consider cool rather than mild*

According to the Köppen climate classification (see Csa and BSh climate type at Peel et al., 2007), Cyprus has a subtropical climate (Mediterranean and semi-arid type) with very **mild winters** and warm to hot summers. As such, we would prefer to keep the proper term (mild) used in the scientific literature to describe the Cyprus winters. The text has been revised to: "short, mild and wet winter vs. long, hot and dry summer"

*Line 97 – use of ca. reconsider throughout text. Circa is not always appropriate (see line 166), and the term 'about' or 'approximately' is modern usage. Historically, it is most commonly used in reference to a date that is not accurately known.*

We have updated the text here and elsewhere in the manuscript. "Ca." is replaced throughout the text with "approximately", "around", "roughly", and "almost". See lines 97, 166, 264, 353, 358, 725.

*Line 99 – c.a. or ca. please harmonise, should be ca. (see comment for line 97)*

See response above. "Ca." has been modified throughout the text.

*Line 127-138 – (i) SOP by the Cost Action COLOSSAL, that should be referenced. (ii) It is assumed that the Q-ACSM operated here had a PM1 lens as cyclone cutoff was 1.3 um, but it should be stated. If you used CDCE, it is a standard vaporiser, but this should also be stated. (iii) Mass concentrations are not calculated via the CDCE, but rather*

*corrected by it. Concentrations are calculated based on signal intensity and Fragmentation Table. (iv) Please also include RIE and RF values used. (v) It is stated RH was below 30%, how was it monitored?*

i.  Standard operating procedure from Cost COLOSSAL Action has been applied, and a reference has been added to line 129 (COST Action CA16109 COLOSSAL Chemical On-Line Composition and Source Apportionment of fine aerosol, Working Group 1. Quadrupole Aerosol Chemical Speciation Monitor (Q-ACSM) - Standard Operating Procedure. Deliverable 1.1. Released in May 2021. https://www.costcolossal.eu/). *The relevant info added is:* 'On-line aerosol instrumentation has been operated following the Standard Operating Procedures defined by ACTRIS (https://www.actris.eu), the European Research Infrastructure on Aerosols, Clouds, and Trace Gases and Cost COLOSSAL (CA16109, 2021).'

ii. A $PM_1$ cyclone and a standard vaporizer have been used. We have updated the text here: "The instrument, along with a scanning mobility particle sizer (SMPS, described below), sampled through a sharp cut cyclone operated at 4 L min-1 (SCC 1.197, BGI Inc., USA), and was equipped with a $PM_1$ aerodynamic lens, yielding an aerosol cut-off diameter of approximately 1.3μm." "The ACSM is designed and built around similar technology as the aerosol mass spectrometer (Jayne et al., 2000), where an aerodynamic particle focusing lens is combined with particle flash vaporization in a high vacuum on the surface of a standard tungsten vaporizer heated at 600 °C, followed by electron impact ionization, separation and final detection of the resulting ions using a quadrupole mass spectrometer".

iii. Indeed there was a typo in that sentence mass concentrations are not calculated via the CDCE, but rather corrected by it. Concentrations are calculated based on signal intensity and Fragmentation Table. The revised sentence below: Mass concentrations are corrected for incomplete detection due to particle bounce using the chemical composition-dependent collection efficiency algorithm (CDCE) (Middlebrook et al., 2012).

iv. We have included the values used in the supplementary document Table S2 and added a new sentence in the manuscript. Instrument response factor (RF) and relative ionization efficiencies (RIE) are reported in table S2.

|  | Value |
| --- | --- |
| $RF_{NO_3}$ | 4.78 *10-11 |
| $RIE_{NH_4}$ | 5.2 |
| $RIE_{SO_4}$ | 0.51 |
| $RIE_{Cl}$ | 1.3 |

v. The efficiency of the nafion dryer placed upstream of the Q-ACSM was checked on a weekly basis using an RH sensor. Since the RH was not monitored continiously the text has been removed from the revised manuscript.

*Line 139 – should read '…were conducted using an 7-wavelength aethalometer (AE33, Magee Scientific, USA) a…'*

Text has been revised accordingly.

*Line 139 -146 – For clarity, what tape was used for the AE33, was it the post 2017 tape (no. 8060) that had non-linearity issues with short wavelengths (https://mageesci.com/tape/Magee_Scientific_Filter_Aethalometer_AE_Tape_Replacement_discussion.pdf )? Not an issue for BC6 but an issue for BB fraction. What was the MAC used?*

The tape used throughout the campaign was part no. 8060, which replaced the "defective" one (part no. 8050). Thus, the absence of non-linearity issues should be expected. Given the above, the c-value was set to 1.39, replacing the old value of 1.57, according to the manufacturer's guideline. MAC values used were the default AE-33 values described by Drinovec et al., 2015.

*Using an AAE of 1 and 2 for FF and WB is the default settings. However, using an AAE of 2 for WB is risky in coastal sites unless WB is the only residential solid fuel burning source. For example, if other fuel types such as turf or peat are used the AAE could be quite different due to the combined effects of combustion efficiency and fuel moisture content while the AAE can also be generally affected by photochemical aging and the mixing state of black carbon (Garg et al. 2016). Additionally, the AAE value can be highly sensitive to small changes at coastal sites. If you apply the Zotter at al. 2017 values for FF and biomass burning (BB rather than WB) of 0.9 and 1.68 rather than 1 and 2, it can lead to a doubling of the BB percentage (Spohn 2021).*

*Can the authors please elaborate on the known fuel sources and why the default values were used rather than optimising the alpha values?*

*Garg, S., Chandra, B. P., Sinha, V., Sarda-Esteve, R., Gros, V., and Sinha, B.: Limitation of the Use of the Absorption Angstrom Exponent for Source Apportionment of Equivalent Black Carbon: a Case Study from the North West Indo-Gangetic Plain, Environmental Science & Technology, 50, 814-824, 10.1021/acs.est.5b03868, 2016.*

*Zotter, P., Herich, H., Gysel, M., El-Haddad, I., Zhang, Y., MoÄ□nik, G., Hüglin, C., Baltensperger, U., Szidat, S., and Prévôt, A. S. H.: Evaluation of the absorption Ångström exponents for traffic and wood burning in the Aethalometer-based source apportionment using radiocarbon measurements of ambient aerosol, Atmos. Chem. Phys., 17, 4229-4249, 10.5194/acp-17-4229-2017, 2017.*

*Spohn, T. K.: A study of black carbon and related measurements from Ireland's atmospheric composition and climate change network, Thesis (Ph.D.): NUI Galway, Galway*

We would like to thank the reviewer for this important comment.

To begin, we need to note that Nicosia is not a coastal site. The city lies at a distance of about 40km from the shore. More specifically, it is located in the middle of a valley

separating two mountain ranges (located to the north and the south) culminating above 1,000m and 1,800m asl, respectively, therefore isolating the agglomeration from coastal influence (e.g., see breeze effects).

In terms of solid fuel used on the island, to the best of our knowledge, based on the latest data available on household typology, coming from the 2011 census in Cyprus and reports by the Cyprus Energy Agency as summarized by the European project EPISCOPE (https://episcope.eu/building-typology/country/cy/), fuel types such as turf, peat, smoky coal etc. should not be expected to be in use for household heating in the country. In fact, for all types of residential buildings (single-family houses, terraced houses or multi-family houses), most heat generators rely on oil, electricity, and gas. Fireplaces are found in 7.8%, 6.7% and 1% of single-family, terraced and multi-family houses, respectively. Other heating methods, which could include stoves or furnaces designed for the fuel types proposed by the Reviewer, seem to account only for 0.05%, 0.001% and 0.005% for the three housing categories as numbered above.

Nevertheless, the AAE associated with wood burning can still vary, depending on the type of wood used and the combustion phase, the particles mixing state and photochemical aging. However, wood in Cyprus shouldn't contain a moisture level as high as it does in Western/Northern Europe due to the much warmer conditions in the region.

Considering the above, and even though such a discussion falls beyond the scope of this study, we have performed a sensitivity analysis on the AAE values used for fossil fuel combustion ($a_{ff}$) and wood burning ($a_{wb}$) used in the aethalometer model. In this context, the aethalometer model was implemented using all the different combinations with $a_{ff}$ varying between 0.8 and 1.2 with an increment of 0.05 and $a_{wb}$ varying from 1.4 through 2.4 with an increment of 0.1. Linear regression was consequently performed between $BC_{wb}$ and the OA concentration at m/z = 60, a fragment directly linked to levoglucosan, thus used as a biomass-burning tracer. Furthermore, linear regression of $BC_{ff}$ vs xylenes ($C_8H_{11}$) was also performed for the cold period, when VOC measurements were available.

[Figure]

Figure 3. Squared Pearson correlation coefficient ($R^2$) for the linear regression of (a) $BC_{wb}$ versus OA at m/z=60 and $BC_{ff}$ versus xylenes, keeping an $a_{ff}$ value of 1 and varying $a_{wb}$.

As depicted in Fig. 3 when keeping an $a_{ff}$ value of 1, no change has been observed for the correlation of $BC_{wb}$ to OA at m/z=60 with varying values for $a_{wb}$, with $R^2$ being 0.822 constantly for $a_{wb}$ above 1.5. For the same scenario, a rather insignificant increase in $R^2$ values was observed for the correlation of $BC_{ff}$ to xylenes when moving from $a_{wb}$=2 to $a_{wb}$=2.1. Furthermore, applying similar values for the AAE to the ones proposed by Zotter et al., 2017, i.e., $a_{ff}$=0.9 and $a_{wb}$=1.7, the $R^2$ value for the $BC_{wb}$ vs OA at m/z=60 correlation was lower ($R^2$=0.803).

Finally, we need to note that the correlation between daily averaged $BC_{wb}$ values and concurrent levoglucosan concentrations, a well-established biomass burning tracer, obtained through filter sampling, is excellent ($R^2$ = 0.94) when using the "default" AAE values ($a_{ff}$=1 and $a_{wb}$=2). Thus, the selection of these values for the Aethalometer model, seems to provide results that capture the residential wood burning phenomenon in a quite satisfactory manner for the urban background conditions in Nicosia. In the revised manuscript we included the sensitivity analysis for AAE values described here (Section 2.2).

*Line 152 – 154 – Why convert volume concentrations to mass concentrations by assuming variable density? Why not convert ACSM measurements to volume as you know the bulk mass fraction of species? You are adding 1 extra unnecessary approximation. Please try by volume and see if there are any differences.*

The variable density applied in our study is based on the bulk mass fraction of species and the densities of the respective chemical components. The calculation of the volume concentration from ACSM (as proposed by the reviewer) will use the same hypotheses the other way around. It is not clear to us how the comparison will be improved. According to other studies, there are no significant differences when you compare the two approaches (see, for instance, Poulain et., 2020). Note that converting the BC mass concentration (from AE33) into volume concentration (to compare it with SMPS) should also lead to additional uncertainties (the shape of BC is still widely debated). Eventually, SMPS is more commonly used to reconstruct the mass of PM$_1$ and compare it with data from both Q-ACSM and AE33 as in the scientific literature (Chazeau et al.,2021; Bougiatioti et al., 2014; Heikkinen et al., 2020), so it made sense to use this common approach. Our paper is about mass concentration, so it appeared logical to keep it simple and compare mass concentrations between all the different techniques (Q-ACSM, AE33, SMPS, Filter-based chemical constituents).

*Line 142 – C = 1.39 is a correction to C=1.57 from Drinovec et al. 2015.*

*Drinovec, L., MoÄ⬜nik, G., Zotter, P., Prévôt, A. S. H., Ruckstuhl, C., Coz, E., Rupakheti, M., Sciare, J., Müller, T., Wiedensohler, A., and Hansen, A. D. A.: The "dual-spot" Aethalometer: An improved measurement of aerosol black carbon with real-time loading compensation, Atmos. Meas. Tech., 8, 1965–1979, https://doi.org/10.5194/amt-8-1965-2015, 2015.*

Please check the answer above, Line 139 -146.

*Line 174 – please report in asl as you did for the stations rather than above ground.*

The altitude of the meteo station was 164m above sea level and 10m above ground. We would like to keep here the information about the station's installation height, though, to

make clear that the meteorological data used are not affected by local topography and the surrounding built environment. The text has been modified as follows:

Standard meteorological parameters (temperature, relative humidity, wind speed and direction) were obtained from the meteorological station of the Cyprus Department of Meteorology, installed 10 m above ground, located at the Athalassa Forestry Park (164 m asl) lying approximately 1.3 km east of the CAO-NIC station.

*Line 201 – m/z used for the first time and not defined.*

Please note that we have defined the mass-to-charge ratio in line 201.

*Line 216 – consider adding literature for BBOA*

*Lin, C., Ceburnis, D., Hellebust, S., Buckley, P., Wenger, J., Canonaco, F., Prévôt, A. S. H., Huang, R.-J., O'Dowd, C., and Ovadnevaite, J.: Characterization of Primary Organic Aerosol from Domestic Wood, Peat, and Coal Burning in Ireland, Environmental Science & Technology, 51, 10624-10632, 10.1021/acs.est.7b01926, 2017.*

*Trubetskaya, A., Lin, C., Ovadnevaite, J., Ceburnis, D., O'Dowd, C., Leahy, J. J., Monaghan, R. F. D., Johnson, R., Layden, P., and Smith, W.: Study of Emissions from Domestic Solid-Fuel Stove Combustion in Ireland, Energy & Fuels, 35, 4966-4978, 10.1021/acs.energyfuels.0c04148, 2021.*

New literature has been added for the BBOA and the different biomass-burning profiles in various regions around the world. The text now reads: On the other hand, given the BBOA factor's sensitivity to the type of solid fuel used, different biomass-burning factor profiles have been reported in various regions around the world (Xu et al., 2020; Trubetskaya et al., 2021).

Additionally a typo in the following lines  was found and corrected.

- L 213: a constrained BBOA factor with the a-values from 0.2 to 0.5 from NG et al., 2011

- L 214: a constrained cooking OA (COA) factor from Crippa et al., 2014 instead of Mohr et al., 2012

*Line 226 – 46% variation is quite high, are you sure BBOA should be related to BCwb, as discussed wood burning may not be the main source of fuel for home heating. This may also change the AAE value for the non-ff BC.*

As discussed above, domestic heating using biomass burning in Cyprus is expected mostly from wood combustion. An a-value of 0.46 for the highly variable BBOA factor still lies well within the ranges reported in a variety of publications applying the a-value approach around the world, as well as the guidelines proposed by both Crippa et al. (2014) and more recently by Chen et al. (2022), which are more focused in the European setting. In this study, the derived BBOA factor has been compared to all other external variables available and found that had strong correlation with $BC_{wb}$ ($R^2$ = 0.84) and Levoglucosan ($R^2$=0.92). As discussed in chapter S1 of the Supplementary material accompanying the submitted article, a sensitivity analysis for the most stable solution has been performed using a-values from 0 to 0.5 with a step of 0.02. The selected solution, applying an a-value of 0.46, was chosen because of both its repeatability as well as its correlation to external BB tracers. As for the fuel types used in Cyprus for space heating and the AAE value used for the non-ff BC component, please refer to our detailed response above.

Cyprus makes extensive use of electricity for space heating (Koroneos et al., 2005). Wood-burning activities are part of domestic heating. To our knowledge, no other solid fuel is used for domestic heating in Cyprus.

*Line 237 – S4 order – a should come before b*

The order of the S4 figures has been changed according to the appearance in the main text.

*Line 236-239 – Why compare PM and not Volume from the SMPS? (see comment for line 152-154)*

Please check the response in the comment before.

*Line 246-249 – Why does the low ratio clearly denote a major contribution of long-chain hydrocarbon OA, is there the possibility that the Org RIE used is incorrect and the OA concentrations are a bit too low?*

The OA/OC ratio typically relates to the quantity of non-carbon atoms present in Organic Aerosols (i.e. mainly H, O, N). High OA/OC ratios typically relate to OA with high contents of O (atomic mass 16), therefore highly oxidized, while low OA/OC ratios relate to OA with low contents of O, therefore containing H (atomic mass 1), i.e. in the form of $-(CH2)_n-$ -long-chain hydrocarbons) which are usually found in primary (hydrogen-like) OA (e.g. Aiken et al., 2008).

It is true that even though the selected RIE = 1.4 for Organics is a well-established and widely used value, being able to describe OA concentrations in the ambient environment adequately, several studies recently have suggested that the specific OA chemical species and their oxidation state, thus consequently the type of OA sources present in the sample, may affect the OA RIE. It has been suggested that reduced organic aerosol found in fresh aerosol from various combustion-related sources (e.g., HOA, COA, BBOA) may be better quantified by RIE values larger than 1.4, while more oxidized OA was found to be sufficiently quantified using this value (Jimenez et al., 2016; Canagaratna et al., 2007; Xu et al., 2018; Katz et al., 2021). We must note that the RIE lies at the denominator of the ACSM and AMS quantification equation (Ng et al., 2011), meaning that adopting a larger RIE for OA would further decrease the reported concentrations.

Nevertheless, we have toned down the statement in Lines 246 – 249, also adding as a possible explanation for the observed slope, the different size fractions compared in Fig. S4, i.e., $PM_1$ for the ACSM versus $PM_{2.5}$ for the filter sampling.

*Line 279-280 – That is not clear from the figure. It seems the reverse, that in the warm period Cluster 5 originates over Middle East/Asia. Also, several clusters pass over Turkey. Please specifically name the clusters and refer to figure in the main text.*

We would like to thank the reviewer for this comment. We have elaborated on the trajectory cluster analysis in the revised text, describing the clusters in more detail. However, we chose to keep numbering the clusters rather than specifically naming each one to avoid conflicting definitions and naming between the cold and warm periods.

*Figure 3 – no need for the 100th place decimal place on the pie charts. Are the data stacked onto each other in the time series, or overlaid? In other words, are the 6-hr average peaks greater than 50 ug/m3 or do the peaks show total NR-PM1?*

We have used the hundred on the decimal place because $Cl^-$ concentration is so low (especially for the warm period) that if we rounded it off, it would read 0. The data are stacked onto each other, so the peaks present the total $NR-PM_1$ for 6-h average peaks. This has become clearer in the caption of figure 3 in the revised manuscript.

*Line 304-310 – Significant figures are incorrect. All averages are only as accurate as the standard deviation. E.g. 10.30 ± 7.92 should be 10 ± 8. Check significant figures in the numbers throughout the text (E.g. Table 1).*

Out of consistency and as a matter of appearance, we used the same number of digits for both averages and standard deviation.

*Line 317 – omit the word much. SO4 is 3 ± 2 ug/m3 which is within the range reported in other studies shown in Table 2.*

We agree with the reviewer that average $SO_4^{2-}$ concentrations lie close to the ones found in the reported studies, and thus the word "much" is omitted in the revised manuscript.

*Figure 4 – These are diurnal averages, the ranges should be shown. Left and right axis should be the same scale. Correct for all average plots.*

Diurnal plots have been revised and show median diurnal trends with ranges (25th & 75th percentile) for all $PM_1$ species during the cold period (Fig 4a.) and warm period (Fig 4b).

[Figure]

[Figure]

(a)                                                     (b)

Figure 4: Median diurnal trends for all ACSM and AE-33 species during the cold period (a) and warm period (b). Shaded area represents the 25th and 75th percentiles of the diurnals.

*Line 386-387 – First definition of MO-OOA. While MO-OOA is synonymous with low-volatile OOA, it should be first defined here as More Oxidised OOA. Same argument for the first mention of LO-OOA. Suggest changing the text to read more like, '… namely the low-volatile MO-OOA (More-Oxidized Oxygenated Organic Aerosol) and semi-volatile LO-OOA (Less-Oxidized Oxygenated Organic Aerosol)…'*

We thank you for the suggestion. Text has been revised accordingly.

*Figure 10 – Shows PSCF 75th Percentile (should represent the probability of an air parcel to be responsible for measured concentrations at the receptor site above the 75th percentile) - please amend the figure text to reflect what 75th Percentile means in this figure*

The text in each panel was deleted and the figure caption has been rewritten in the revised manuscript.

'PSCF plots for MO-OOA during the cold and warm period. The color scale represents the probability of air parcels arriving at the receptor site (white dot) for measured concentrations higher than the 75th percentile.'

*Line 659 – Figure S19 shows LV-OOA, should it show MO-OOA as the figure caption and text suggest? Rather than abbreviated as low-volatility*

Thank you for spotting this error. There was indeed a typo in the title of the right y-axis, which has been corrected. Now MO-OOA is used everywhere and matches the figure caption and the text.

*Line 702 – insert the word of, '… of the acquired data'*

The text was adapted accordingly.

*Figure S4 – a) should be BC AE33 rather than ACSM*

The typo in the y-axis title of the figure has been corrected.

*Figure S6 – a) cluster 1 and 7 are difficult to differentiate – why has 7 clusters been chosen rather than another number? Is warm and cold switched in the text – refer here to comment about line 279-280.*

Choosing the number of clusters used in each season was the result of combining information on the percentage change in Total Spatial Variance (TSV) as a function of the number of clusters of merged trajectories and an empirical evaluation of the resulting mean trajectory paths of each cluster.

[Figure]

Figure 5. Percentage change in Total Spatial Variance as a function of the number of clusters for the (a) cold and (b) warm periods respectively.

When looking at the cold period TSV (Fig. 5a), the sharpest decline is observed when moving from three to four clusters. However, four clusters could not quite represent all the recorded trajectories and especially the ones describing air masses arriving in Nicosia from the east. Thus, seven clusters were selected, first of all since that's where the next significant drop in TSV is observed, but also because with seven clusters, the eastern-bound air masses are represented in a more satisfactory manner. The same reasoning was used for the warm period (Fig. 5b). After the large drops in TSV for the two – to – three and the three – to – four cluster transition, the next quite significant decrease was recorded when moving to seven clusters. As mentioned earlier, these details are provided in the revised main text (Section 3.2). At the same time, the TSV plots have also been included in the supplement.

*Figure S9 – it would be good to insert the correlations graphs between the factors in warm and cold periods. E.g. HOA-1 (cold) vs HOA-1 (warm).*

The correlation graphs between the factors in the two seasons can be found in graphs S10(a-e). E.g., $R^2$ for HOA-1$_{cold}$ and HOA-1$_{warm,}$ according to the correlation matrix, is 0.99 (bottom right corner of figure S10a).

*Figure S17 – Also include the Log10(n+1) trajectory footprint graphs from Zefir for the PSCF.*

We have included the Log10(n+1) trajectory density plots (Fig. 6) for both the cold and warm periods in Figure S17 in the supplement.

[Figure]

Figure 6. Log10(n+1) trajectory density plot for both the cold and warm periods

*Figure S19 – rename MO-OOA rather than LV-OOA*

Please see the response above. LV-OOA is replaced overall by MO-OOA.

**Reviewer 2**

*This study is a very nice and detailed work that indeed was lacking for the specific area. The manuscript contains a lot of interesting information, and it should be published. However, it is necessary to polish some parts of it.*

We would like to thank the Reviewer for their positive assessment of our work as well as for providing the constructive comments below, aiming at improving the manuscript and making the objectives as well as the concluding arguments even clearer. We will attempt to address each one in our point-by-point response.

1. *The title denotes that the main subject of this paper is the transported pollution from Middle East to Cyprus. However, backtrajectories indicated that only a small contribution was from Middle East and that the main origin of the air parcels arriving in Nicosia were originated from Europe. Thus, regional pollution from Cyprus and long-range transported pollution from Europe is much more important. Middle East had rather a minor contribution to the $PM_1$ levels measured in Cyprus. Thus, the corresponding parts in the manuscript should be revised.*
2. *As a consequence, the title of the paper should be changed.*

We would like to thank the Reviewer for raising this point, thus allowing us to make our argument clearer for the reader in the revised version of the manuscript. Indeed, a similar point has been raised by Reviewer #1. Given the relevance of points #1 and #2, we will address both in a single response.

In this direction, we have provided significantly more detail in the section of the manuscript describing the air mass back trajectory clustering analysis, arguing that a significant portion of air masses arriving at the measuring site either originates or passes above the Middle East. This aspect is more pronounced during the Cold period when 25% of the calculated trajectories are assigned to Cluster 1, the mean path of which is clearly associated with the Middle East.

Our assessment, though, regarding long-range transported pollution arriving on the island is not, first and foremost, based on the clustering analysis provided. As shown by the PSCF analysis performed for both measured $SO_4^{2-}$ as well as the more oxidized organic aerosol component (MO-OOA), higher concentrations for both species (above the 75$^{th}$ percentile) were found to be associated with air masses spending time over regions in Turkey and the Middle East for the cold and warm periods. We need to note that MO-OOA contributed almost half of OA during both the cold and warm seasons (44% and 45%), while $SO_4^{2-}$ accounted for 23% and 35% of $PM_1$ for the two studied periods, respectively. Adding to the above, we need to mention the documented – through NWR – relationship of even the "more primary" species (i.e., $BC_{ff}$ and HOA-2) as well as the LO-OOA component with elevated wind directions of an Eastern origin, that has also been extensively discussed in the manuscript. We thus feel that there is merit to our concluding remark that a significant portion of $PM_1$ aerosol over Nicosia can be linked to transported air masses from Turkey and the Middle East. However, trying to address both the Reviewers' concerns, we have revised the manuscript title to:

"Ambient carbonaceous aerosol levels in Cyprus and the role of pollution transport from the Middle East. "

*3. Regarding the inorganic species: using the mass fractions in Figure 3 or the concentrations in Table 1 in order to check the mass balance of $SO_4^{-2}$, $NH_4^+$ and $NO_3^-$ one will expect to see neutralized ammonium sulfate $((NH_4)_2SO_4)$ particles plus ammonium nitrate particles $(NH_3NO_4)$.*

*However, for the cold period even if we assume that the sulfate particles are not fully neutralized (existing in the form of $NH_4HSO_4$ or $NH_4HSO_4 + (NH_4)_2SO_4$), the remaining $NH_4$ is quite low (one order of magnitude lower compared to the $NO_3$ amount) and only a very small amount of $NH_4NO_3$ could be justified. As a consequence, there is not enough $NH_4$ to form $NH_4Cl$. In what form could be the rest of $NO_3$ and the $Cl^-$? The diurnal profile of $NO_3$ (Figure 4) seems very similar to the organics. Could it be organonitrates? Could it originate from BBOA? Could it be $KNO_3$? Is it possible for the $Cl^-$ to be in the form of KCl? Could part of the $SO_4^{-2}$ be in the form of $K_2SO_4$? According to the manuscript, the 24 h*

*filters were analyzed for $K^+$ and $Cl^-$ (among other ions). How well does $K^+$ correlate to $NO_3^-$? Or to $Cl^-$?*

*During the warm period ammonium is not enough to neutralize the sulfate. Again, if the sulfate particles are not fully neutralized (existing in the form of $NH_4HSO_4$ or $NH_4HSO_4$ + $(NH_4)_2SO_4$) the remaining $NH_4$ could make just a small amount of $NH_4NO_3$. In what form could be the rest amount of $NO_3$ given the fact that during this period there is no biomass burning sources?*

*Please explain, discuss and revise the above in the manuscript (e.g., lines 302-303, 370-378, 457).*

*ACSM tends to overestimate $NO_3^-$ concentrations in comparison to the filters (according to Figure S4d), but $NH_4^+$ has much better correlation, so it cannot be $NH_4NO_3$ that evaporates from filter (please revise lines 242-243).*

The neutralization was studied for both online and offline inorganic measurements. The equation nr. 3 from Jiang et al. (2019) was used to calculate the predicted $NH_4^+$.

$$NH_{4Predicted}^+ = 18 \times \left( 2 \times \frac{SO_4^{2-}}{96} + \frac{NO_3^-}{62} + \frac{Cl^-}{35.5} \right) \ (Equation \ Nr \ 3. Jiang \ et \ al. (2019)$$

The results denote that inorganic species measured by ACSM for both seasons have a slope between measured and predicted of 0.8 during the cold period (Fig. 7a) and a slope of 0.78 for the warm period (Fig. 7b).

[Figure]

Figure 7:    NH$_4^+$ measured vs. NH$_4^+$ predicted for the cold period (a) and warm period (b).

This result suggests that 20% of the predicted ammonium is unexplained. According to Crenn et al. (2015), these measurements fall between the uncertainty of ammonium and/or sulfate concentrations estimated during the largest Q-ACSM intercomparison study. In this study, the reproducibility uncertainties for NR-PM$_1$ was assessed, and it showed an uncertainty of 15, 28 and 36% for nitrate, sulfate and ammonium, respectively.

For our filter-based off-line neutralization test, the results showed a slope close to one (0.96) between measured and predicted NH$_4^+$ supporting a full neutralization of sulfate and nitrate by ammonium (Fig. 8).

[Figure]

Figure 8: NH$_4^+$ measured vs. NH$_4^+$ for filter samples.

Note that this result is aligned with the full neutralization systematically recorded from filter-based $PM_{2.5}$ chemical analysis performed every year at a Nicosia urban background (annual member state report submitted to the European Environmental Agency by the national air quality network in the framework of EU air quality directives).

In conclusion, our results suggest a full neutralization so that alternative hypotheses (as proposed by the reviewer) do not need to be further explored in the paper. A short note has been added to the revised version to account for the full neutralization.

*4. MO-OOA during the cold months: It is very difficult to transform fresh OA into that highly OOA within 4 hours during nighttime. Kodros et al. (2021) showed that it is possible to transform fresh BBOA in OOA (in dark aging experiments) within 3 hours resulting in OOA with a m/z 44 around 0.13. According to Figure S9a the MO-OOA factor has an m/z 44 around 0.3, which corresponds to much more oxygenated compounds. It would be better to remove the sentence in lines 490-491 unless it can be supported by a reference from literature. Nevertheless, there should be an explanation in the manuscript for the increase of the MO-OOA during the night.*

*In addition, MO-OOA seems highly oxidized (m/z 44 ~0.3) even though it refers to cold months. OOA factors during the winter usually are less oxidized (m/z 0.1-0.15) (e.g., Crippa et al.,2013; Florou et al., 2017). How is this behavior explained? Is it real or is it due to the way the PMF was run (i.e., two factors were constrained). If you use as a constrain solution a winter OOA mass spectrum, how do the rest mass spectra (LO-OOA and HOA-2) change? Is there still the increase of the MO-OOA during the night?*

Crenn et al. (2015), during an intercomparison study of several Q-ACSM instruments, reported significant variability for the mass contribution of m/z = 44 to total OA ($f_{44}$), among thirteen different ACSMs being part of the exercise. Based on the same dataset, Frohlich et al. (2015), going in more depth for this specific matter, reports, that even though $f_{44}$ absolute values may vary, the ratios of $f_{44}$ between different instruments, even among those exhibiting the highest differences in absolute numbers remained stable. Keeping that in mind, the same study argues that OA source apportionment is not compromised

by this aspect. In fact, performing an ME-2 PMF to all datasets, using a unified approach, concluded that the instrument-related $f_{44}$ variability, whilst being transferred to the respective deconvolved OOA mass spectra $f_{44}$, yielding a variability in OOA source profiles for different instruments, does not affect the relatively small discrepancies of the contributions appended to each OOA factor. The $f_{44}$ in OOA factor profiles has been reported to be, in general, larger in the ACSM as opposed to the AMS (Ng et al., 2011), while a mechanism controlling such behavior, has been already suggested (Pieber et al., 2016), involving semi-refractory residues of carbonaceous species reacting with thermal decomposition products of inorganic salts, on the heated vaporizer surface, inducing signal at m/z = 44, not related to sampled OA, with this interference being highly variable depending on instrument and it's field deployment history.

In view of the above, it would seem that direct comparison of the $f_{44}$ among different instruments, either within the bulk OA mass spectra or within a ME-2 deconvolved OOA factor, should be exercised with quite some caution, if not avoided (Frohlich et al., 2015). On the other hand, comparisons for quantified PMF factor concentrations and contributions to total OA seem to be more straightforward, of course keeping in mind the overall uncertainties of both measuring OA via aerosol mass spectrometry as well as those induced by the PMF model itself.

It needs to be noted that wintertime MO-OOA factors with $f_{44}$ values comparable to the ones deconvolved in this study have been also reported elsewhere in the literature. For example, Zhang et al. (2019), in a six-year seasonal OA source apportionment in Paris, France, reports MO-OOA factors for each of the monitored winters, exhibiting $f_{44}$ values reaching or even exceeding 0.3 on occasion. In addition, remaining in the northern European setting during wintertime, Lin et al. (2022) report a MO-OOA factor with $f_{44}$ of around 0.3 for the city of Galway in Ireland. It is worth mentioning that both the above studies have reported enhanced MO-OOA concentrations during nighttime in the winter, with both arguing that such enhancement is related to nighttime atmospheric processing of locally emitted OA. Similar findings, in both terms of high $f_{44}$ values and nighttime enhancement during wintertime, have also been reported for the Eastern Mediterranean

urban environment (Athens, Greece), where a link to oxidized primary residential wood burning emissions as a potential driver of the low volatility OOA factor diurnal variability, was also suggested (Stavroulas et al., 2019).

In this context, even though we decided to keep the sentence in Lines 490-491, the aforementioned references and a bit more discussion are added in the revised manuscript, trying to address the Reviewers' concerns. Finally, following the guidelines for running ME-2 PMF as provided in several relevant publications (Crippa et al., 2014; Frohlich et al., 2015; Chen et al., 2022), we chose to apply no constraints for the secondary OA factors when running the model. The sensitivity analysis the Reviewer is suggesting, we feel, would stumble upon important problems, first and foremost derived by the anchor profile selection itself – given the $f_{44}$ variability discussed earlier that could compromise the anchor profile representativeness – and secondarily related to the optimization of the a-value applied.

A few sentences and references have been added in the revised manuscript to account for the above (Section 3.5).

*5. Mass spectra comparison (Figure S10): ASCM and Q-AMS mass spectra are usually manipulated using the old fragmentation table of Allan et al. (2007), while the HR-AMS has incorporated the new fragmentation table of Aiken et al. (2009). This means that m/z 18 and 28 are calculated differently. In Figure S10 some of the compared mass spectra have been treated with the old fragmentation table and some of them with the new fragmentation table. When comparing mass spectra derived by different fragmentation tables, were m/z's 18 and 28 excluded or not? This must be clarified in the text.*

Response: First, m/z 28 was not part of the comparison to literature mass spectra since the $N_2$ signal dominates this m/z and is excluded even from the PMF input matrix. Regarding m/z 18, we followed the reviewer's correction to exclude it from the comparison and have updated the correlation plots in Supplementary Figure S10a-e. As the Reviewer may see, there are no significant changes in the correlation diagrams.

*6. Figure S9 is important, and I suggest it should be transferred to the main paper.*

We thank the reviewer for the suggestion. We moved the factor profiles figure in the main manuscript.